# HYPERPARAMETER OPTIMIZATION THROUGH NEURAL NETWORK PARTITIONING

**Bruno Mlodozeniec**[†*]**, Matthias Reisser**[‡]**, Christos Louizos**[‡]
[†]University of Cambridge, [‡]Qualcomm AI Research
`bkm28@cam.ac.uk`, {`mreisser,clouizos`}`@qti.qualcomm.com`

## ABSTRACT

Well-tuned hyperparameters are crucial for obtaining good generalization behavior in neural networks. They can enforce appropriate inductive biases, regularize the model and improve performance — especially in the presence of limited data. In this work, we propose a simple and efficient way for optimizing hyperparameters inspired by the marginal likelihood, an optimization objective that requires no validation data. Our method partitions the training data and a neural network model into $K$ data shards and parameter partitions, respectively. Each partition is associated with and optimized only on specific data shards. Combining these partitions into subnetworks allows us to define the "out-of-training-sample" loss of a subnetwork, *i.e.*, the loss on data shards unseen by the subnetwork, as the objective for hyperparameter optimization. We demonstrate that we can apply this objective to optimize a variety of different hyperparameters in a single training run while being significantly computationally cheaper than alternative methods aiming to optimize the marginal likelihood for neural networks. Lastly, we also focus on optimizing hyperparameters in federated learning, where retraining and cross-validation are particularly challenging.

## 1 INTRODUCTION

Due to their remarkable generalization capabilities, deep neural networks have become the de-facto models for a wide range of complex tasks. Combining large models, large-enough datasets, and sufficient computing capabilities enable researchers to train powerful models through gradient descent. Regardless of the data regime, however, the choice of hyperparameters — such as neural architecture, data augmentation strategies, regularization, or which optimizer to choose — plays a crucial role in the final model's generalization capabilities. Hyperparameters allow encoding good inductive biases that effectively constrain the models' hypothesis space (*e.g.*, convolutions for vision tasks), speed up learning, or prevent overfitting in the case of limited data. Whereas gradient descent enables the tuning of model parameters, accessing hyperparameter gradients is more complicated.

The traditional and general way to optimize hyperparameters operates as follows; **1)** partition the dataset into training and validation data[1], **2)** pick a set of hyperparameters and optimize the model on the training data, **3)** measure the performance of the model on the validation data and finally **4)** use the validation metric as a way to score models or perform search over the space of hyperparameters. This approach inherently requires training multiple models and consequently requires spending resources on models that will be discarded. Furthermore, traditional tuning requires a validation set since optimizing the hyperparameters on the training set alone cannot identify the right inductive biases. A canonical example is data augmentations — they are not expected to improve training set performance, but they greatly help with generalization. In the low data regime, defining a validation set that cannot be used for tuning model parameters is undesirable. Picking the right amount of validation data is a hyperparameter in itself. The conventional rule of thumb to use $\sim 10\%$ of all data can result in significant overfitting, as pointed out by Lorraine et al. (2019) , when one has a sufficiently large number of hyperparameters to tune. Furthermore, a validation set can be challenging

---

[*]Work done while at Qualcomm AI Research. Qualcomm AI Research is an initiative of Qualcomm Technologies, Inc. and/or its subsidiaries.

[1]a third partition, the test or holdout set is used to estimate the final model performance

to obtain in many use cases. An example is Federated Learning (FL) (McMahan et al., 2017), which we specifically consider in our experimental section. In FL, each extra training run (for, *e.g.*, a specific hyperparameter setting) comes with additional, non-trivial costs.

Different approaches have been proposed in order to address these challenges. Some schemes optimize hyperparameters during a single training run by making the hyperparameters part of the model (*e.g.*, learning dropout rates with concrete dropout (Gal et al., 2017), learning architectures with DARTs (Liu et al., 2018) and learning data-augmentations with schemes as in Benton et al. (2020); van der Wilk et al. (2018)). In cases where the model does not depend on the hyperparameters directly but only indirectly through their effect on the value of the final parameters (through optimization), schemes for differentiating through the training procedures have been proposed, such as Lorraine et al. (2019). Another way of optimizing hyperparameters without a validation set is through the canonical view on model selection (and hence hyperparameter optimization) through the Bayesian lens; the concept of optimizing the *marginal likelihood*. For deep neural networks, however, the marginal likelihood is difficult to compute. Prior works have therefore developed various approximations for its use in deep learning models and used those to optimize hyperparameters in deep learning, such as those of data augmentation (Schwöbel et al., 2021; Immer et al., 2022). Still, however, these come at a significant added computational expense and do not scale to larger deep learning problems.

This paper presents a novel approach to hyperparameter optimization, inspired by the marginal likelihood, that only requires a single training run and no validation set. Our method is more scalable than previous works that rely on marginal likelihood and Laplace approximations (which require computing or inverting a Hessian (Immer et al., 2021)) and is broadly applicable to any hierarchical modelling setup.

## 2 Marginal Likelihood and prior work

In Bayesian inference, the rules of probability dictate how any unknown, such as parameters $\boldsymbol{w}$ or hyperparameters $\psi$, should be determined given observed data $\mathcal{D}$. Let $p(\boldsymbol{w})$ be a prior over $\boldsymbol{w}$ and $p(\mathcal{D}|\boldsymbol{w}, \psi)$ be a likelihood for $\mathcal{D}$ with $\psi$ being the hyperparameters. We are then interested in the posterior given the data $p(\boldsymbol{w}|\mathcal{D}, \psi) = p(\mathcal{D}|\boldsymbol{w}, \psi)p(\boldsymbol{w})/p(\mathcal{D}|\psi)$. The denominator term $p(\mathcal{D}|\psi)$ is known as the *marginal likelihood*, as it measures the probability of observing the data given $\psi$, irrespective of the value of $\boldsymbol{w}$: $p(\mathcal{D}|\psi) = \int p(\boldsymbol{w})p(\mathcal{D}|\boldsymbol{w}, \psi)d\boldsymbol{w}$.

Marginal likelihood has many desirable properties that make it a good criterion for model selection and hyperparameter optimization. It intuitively implements the essence of Occam's Razor principle (MacKay, 2003, § 28). In the PAC-Bayesian literature, it has been shown that higher marginal likelihood gives tighter frequentist upper bounds on the generalization performance of a given model class (McAllester, 1998; Germain et al., 2016). It also has close links to cross-validation (see section 2.1) and can be computed from the training data alone. However, computation of the marginal likelihood in deep learning models is usually prohibitively expensive and many recent works have proposed schemes to approximate the marginal likelihood for differentiable model selection (Lyle et al., 2020; Immer et al., 2021; 2022; Schwöbel et al., 2021).

### 2.1 "Learning speed" perspective

Lyle et al. (2020); Fong and Holmes (2020) pointed out the correspondence between "learning speed" and marginal likelihood. Namely, the marginal likelihood of the data $\mathcal{D}$ conditioned on some hyperparameters $\psi$ can be written as:

$$\log p(\mathcal{D}|\psi) = \sum_k \log \mathbb{E}_{p(\boldsymbol{w}|\mathcal{D}_{1:k-1}, \psi)} \left[ p(\mathcal{D}_k|\boldsymbol{w}, \psi) \right] \geq \sum_k \mathbb{E}_{p(\boldsymbol{w}|\mathcal{D}_{1:k-1}, \psi)} \left[ \log p(\mathcal{D}_k|\boldsymbol{w}, \psi) \right] \quad (1)$$

where $(\mathcal{D}_1, \ldots, \mathcal{D}_C)$ is an arbitrary partitioning of the training dataset $\mathcal{D}$ into $C$ shards or chunks[2], and $p(\boldsymbol{w}|\mathcal{D}_{1:k}, \psi)$ is the posterior over parameters of a function $f_{\boldsymbol{w}} : \mathcal{X} \to \mathcal{Y}$, from the input domain $\mathcal{X}$ to the target domain $\mathcal{Y}$ after seeing data in shards 1 through $k$. The right-hand side can be interpreted as a type of cross-validation in which we fix an ordering over the shards and measure the "validation" performance on each shard $\mathcal{D}_k$ using a model trained on the preceding shards $\mathcal{D}_{1:k-1}$.

---

[2]We use the terms "chunk" and "shard" interchangeably.

Alternatively, it can be viewed as the *learning speed* of a (probabilistic) model: *i.e.*, a measure of how quickly it learns to perform well on new shards of data after only having been fit to the previous shards (through exact Bayesian updating).

This perspective neatly illustrates why models with higher marginal likelihood can exhibit good inductive biases, *e.g.*, encoded through $\psi$, $\boldsymbol{w}$ and $f_{\boldsymbol{w}}$. Namely, such models can be expected to learn faster and generalize better after seeing fewer samples. For example, if the hypothesis space is constrained[3] to functions satisfying symmetries present in the data, we need fewer data to identify the correct function (Sokolic et al., 2017; Sannai et al., 2021). We argue that the "learning speed" aspect of marginal likelihood — *i.e.*, measuring how well the model generalizes to new data in the training set, having been trained only on the previous data points — is the key property making marginal likelihood a useful tool for selecting hyperparameters.

## 2.2 TRAINING SPEED FOR HYPERPARAMETER OPTIMIZATION

Computing the "learning speed", requires samples from the posterior $p(\boldsymbol{w}|\mathcal{D}_{1:k}, \psi)$. Unfortunately, in deep learning settings, such samples are impractical to obtain; thus, prior works have focused on more scalable alternatives. Lyle et al. (2020) propose to approximate the objective in Eq. 1 by looking at the *training speed* during standard training of a neural network by SGD. Specifically, they define the training speed as the reduction in the training loss after a single SGD parameter update, summed over all updates in the first epoch. They argue that, during the first epoch of training, after the neural network parameters, $\boldsymbol{w}$, have been updated with SGD steps using data from shards $\mathcal{D}_{1:k}$, they can be approximately used in place of the sample from the posterior $p(\boldsymbol{w}|\mathcal{D}_{1:k}, \psi)$ in Eq. 1. They extend the analogy to training past one epoch and use the training speed estimate for model selection (Ru et al., 2021). As pointed out by the authors, however, the analogy between learning speed and training speed somewhat breaks down after 1 epoch of training. The network parameters have "seen" every datapoint in the training set after 1 epoch, and hence the connection to measuring the model's generalization capability is weakened.

For the sake of scalability and alignment with deep learning practice, we also focus on simple pointwise approximations $q_k(\boldsymbol{w}) = \delta(\boldsymbol{w} = \hat{\boldsymbol{w}}_k)$ to the posteriors $p(\boldsymbol{w}|\mathcal{D}_{1:k}, \psi)$. However, in contrast to prior work, we explicitly parametrize the learning procedure such that, at any given training iteration, we have access to a model that is trained only on a subset of the data $\mathcal{D}_{1:k}$. In doing so, we can approximate the objective in Eq. 1, and thus use it to optimize the hyperparameters during the entire training run.

## 3 PARTITIONED NEURAL NETWORKS

Our goal is to optimize the objective

$$\mathcal{L}_{\mathrm{ML}}(\mathcal{D}, \psi) = \sum_{k=1}^{C} \mathbb{E}_{q_{k-1}(\boldsymbol{w})} \left[ \log p(\mathcal{D}_k | \boldsymbol{w}, \psi) \right] \tag{2}$$

wrt. $\psi$, which is an approximation to the lower-bound presented in Eq. 1 above. In Appendix A, we show that the left-hand side is also a lower-bound on the marginal likelihood under some unobtrusive conditions. As mentioned in Section 2.2, our goal is to propose an architecture and a training scheme so that we can easily obtain models trained on only subsets of the data $\mathcal{D}_{1:k}$ for all $k$ throughout training. We propose that each $\{q_k(\boldsymbol{w})\}_{k=1}^{C}$ optimizes a subset of the parameters of the neural network, in a manner that allows us to extract "subnetworks" from the main network that have been trained on specific chunks of data. We describe the partitioning scheme below.

**Partitioning the parameters** Denote the concatenations of the weights of a neural network $\boldsymbol{w} \in \mathbb{R}^N$. We can define a partitioning $((\boldsymbol{w}_1, \ldots, \boldsymbol{w}_C), P)$ of the parameters into $C$ partitions, such that $\boldsymbol{w} = P \operatorname{concat}(\boldsymbol{w}_1, \ldots, \boldsymbol{w}_C)$ for a permutation matrix $P \in \{0, 1\}^{N \times N}$. For ease of exposition, we drop the dependence on $P$, assuming that $\boldsymbol{w}$ is already arranged such that $P$ is identity, $P = I_{N \times N}$.

Given the partitioning $(\boldsymbol{w}_1, \ldots, \boldsymbol{w}_C)$ of the parameters, we then specify $C$ subnetworks with weights $\boldsymbol{w}_s^{(1)}, \ldots, \boldsymbol{w}_s^{(C)}$ such that $\boldsymbol{w}_s^{(k)} = \operatorname{concat}(\boldsymbol{w}_1, \ldots, \boldsymbol{w}_k, \hat{\boldsymbol{w}}_{k+1}, \ldots, \hat{\boldsymbol{w}}_C)$, where $\hat{\boldsymbol{w}}_i$ are some default

---

[3] or if the learning algorithm is heavily biased towards returning hypotheses that satisfy a given invariance, *e.g.*, through the use of a prior.

values not optimized during training[4]. More specifically, the $k$-th subnetwork, $\boldsymbol{w}_s^k$, retains the first $k$ partitions from the weight partitioning and sets the remaining parameters to $\hat{\boldsymbol{w}}_{k+1:C}$. Note that, if each $\boldsymbol{w}_k$ is only updated on chunks $\mathcal{D}_{1:k}$, the subnetwork $\boldsymbol{w}_s^{(k)}$ is only comprised of weights that have been updated on $\mathcal{D}_{1:k}$. Thus, we can view the parameters of $\boldsymbol{w}_s^{(k)}$ as an approximation to $q_k(\boldsymbol{w})$. Although, given that a subset of the parameters in each $\boldsymbol{w}_s^{(k)}$ is fixed, this would likely be a poor approximation to the true posterior over the weights given $\mathcal{D}_{1:k}$, it could be, intuitively, a reasonable approximation in function space[5].

**Partitioned training** Having partitioned the dataset $\mathcal{D}$ into $C$ chunks $(\mathcal{D}_1, \ldots, \mathcal{D}_k)$, we update each partition $\boldsymbol{w}_k$ by optimising the negative log-likelihood[6] on chunks $\mathcal{D}_{1:k}$ using subnetwork $\boldsymbol{w}_s^{(k)}$ by computing the following gradients:

$$\nabla_{\boldsymbol{w}_k} \mathcal{L}\left(\mathcal{D}_{1:k}, \boldsymbol{w}_s^{(k)}\right) = \sum_{(\boldsymbol{x},y) \in \mathcal{D}_{1:k}} \nabla_{\boldsymbol{w}_k} \log p\left(y \middle| \boldsymbol{x}; \boldsymbol{w}_s^{(k)}, \psi\right). \tag{3}$$

We interleave stochastic gradient updates of each partition of the weights with updating the hyperparameters $\psi$ using $\mathcal{L}_{\mathrm{ML}}$ in Eq. 2:

$$\nabla_\psi \mathcal{L}_{\mathrm{ML}}\left(\mathcal{D}, \psi\right) \approx \sum_{k=2}^{C} \sum_{(\boldsymbol{x},y) \in \mathcal{D}_k} \nabla_\psi \log p\left(y \middle| \boldsymbol{x}, \boldsymbol{w}_s^{(k-1)}, \psi\right). \tag{4}$$

This can be seen as the sum of the *out-of-sample* losses for each subnetwork $\boldsymbol{w}_s^{(k)}$. The scheme is illustrated in Figure 1. For details of how the updates are scheduled in our experiments, see Appendix I. Note that, while we could incorporate the gradient of the first term from Eq. 1 corresponding to $\mathbb{E}_{q_0(\boldsymbol{w})}[\log p(\mathcal{D}_1 | \boldsymbol{w}, \psi)]$ in Eq. 4, we chose to leave it out. Hence, the gradient of Eq. 4 is of an estimate that can be viewed as an approximation to the *conditional* marginal likelihood $\log p\left(\mathcal{D}_{2:C} | \mathcal{D}_1, \psi\right)$. Conditional marginal likelihood has been shown to have many desirable properties for model selection and, in many cases, can be a better proxy for generalization (Lotfi et al., 2022).

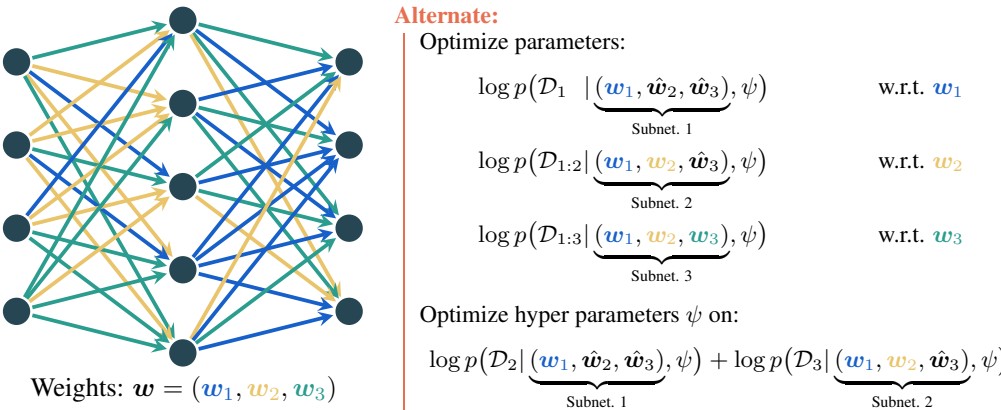

Figure 1: Best viewed in colour. Illustration of the partitioning scheme for a single hidden layer perceptron with $C = 3$ chunks.

This procedure, inspired by the marginal likelihood, has several desirable properties compared to prior work. **1)** Our objective is computationally efficient, with a computational cost roughly corresponding to evaluating subnetworks on the training set. There is no need to compute nor invert a Hessian with

---

[4] *e.g.*, $\hat{\boldsymbol{w}}_i$ could be the value of the weights at initialization, or $\hat{\boldsymbol{w}}_i = \boldsymbol{0}$ corresponding to pruning those parameters and obtaining a proper subnetwork.

[5] Since a) the mapping from parameters to functions is not bijective and b) neural networks are highly overparameterised and can be heavily pruned while retaining performance (Frankle and Carbin, 2018), obtaining a good fit to a subset of the training data with a subset of the model parameters should be possible. Furthermore, "scaling laws" indicate that the benefit of having more parameters becomes apparent mostly for larger dataset sizes (Kaplan et al., 2020), thus it is reasonable for subnetworks fit to more data to have more learnable parameters.

[6] Optionally with an added negative log-prior regularization term $\log p(\boldsymbol{w}_s^{(k)})$.

respect to the weights, as in the Laplace approximation (Immer et al., 2021; 2022). **2)** Our objective is readily amenable to optimization by stochastic gradient descent; we do not have to iterate over the entire training set to compute a single gradient update for the hyperparameters. **3)** Compared to the training speed objective (Lyle et al., 2020), in our method, the training of the weights in each subnetwork progresses independently of the data in future chunks. Hence, it can be seen as more truthfully measuring the generalization capability of a model using a given set of hyperparameters.

**Partitioning Schemes** There are several ways in which the neural network weights can be partitioned. In our experiments in Section 5, we partition the weights before beginning training by assigning a fixed proportion of weights in each layer to a given partition at random. For each subnetwork, for the weight partitions corresponding to future chunks, we use the values of the weights at initialisation. For a discussion of partitioning schemes, see Appendix C.

## 4   RELATED WORKS

**Hyperparameter optimization in deep learning**   Many works have tackled the challenge of optimizing hyperparameters in deep learning. Works on implicit differentiation, such as the one by Lorraine et al. (2019), allow for optimizing training hyperparameters such as the learning rate, weight-decay, or other hyperparameters that affect the final neural network weights only through the training routine. Other works have proposed ways to parameterize and optimize data-augmentations (Cubuk et al., 2018; Li et al., 2020), search-spaces for neural network architectures, as well as methods to optimize architectures using gradient-based optimization (Liu et al., 2018; Elsken et al., 2019). All of the above works have primarily relied on optimizing hyperparameters on a separate validation set and are compatible with the objective defined in this work. Several works have also aimed to cast learning data augmentations as an invariance learning problem. They do so by parameterizing the model itself with data augmentations, and frame invariance learning as a model selection problem (van der Wilk et al., 2018; Benton et al., 2020; Schwöbel et al., 2021; Nabarro et al., 2022; Immer et al., 2022). We compare against Benton et al. (2020) ("Augerino") and Immer et al. (2022) ("Differentiable Laplace") on this task in the experimental section.

**Hyperparameter optimization without a validation set**   A limited number of works consider learning hyperparameters without a validation set in a deep learning context. Benton et al. (2020) propose a simple method for learning invariances without a validation set by regularising invariance hyperparameters to those resulting in higher invariance. They show that the invariances found tend to be insensitive to the regularisation strength, determined by another hyperparameter. However, the method relies on being able to *a priori* define which hyperparameters lead to higher invariance through a suitable regularisation function. In more complex invariance learning settings, defining the regulariser can be challenging. For example, if data-augmentation transformations were to be parameterized by a neural network (as proposed in Lorraine et al. (2019)), it is non-trivial to devise an adequate regulariser. We show that our method can be applied to such settings.

Other works focus on deriving tractable approximations to the marginal likelihood for deep neural networks. Schwöbel et al. (2021) propose only marginalising-out the parameters in the last layer of the neural network by switching it out for a Gaussian Process. They treat the preceding layer effectively as a hyperparameter, and optimize invariance parameters using the marginal likelihood. Although they show promising results on MNIST, they found they "were unable to learn invariances for CIFAR-10" (Schwöbel et al., 2021, §7) and highlighted the need to marginalise lower layers as well. In contrast, our objective can be seen as being inspired by marginal likelihood where arbitrary network layers can be "marginalised", and works on datasets like CIFAR-10. Immer et al. (2022) have adapted the Laplace approximation (Immer et al., 2021) to make it tractable for learning data augmentations. In contrast to Schwöbel et al. (2021), they approximately marginalize out all the network parameters, and performs favourably. Their approximation, however, requires approximations to a Hessian w.r.t. all network parameters; for that reason, their work reports results for architectures only up to a ResNet-14, whereas our method can easily scale to larger architectures.

**Hyperparameter optimization in FL**   Improving hyperparameter optimization is especially relevant to FL. Given the potential system level constraints (Wang et al., 2021), methods that optimize the hyperparameters and parameters in a single training run are preferred. On this note, Khodak et al. (2021) introduced FedEx and showed that it can successfully optimize the client optimizer

hyperparameters. FedEx relies on a training/validation split on the client level and uses a REIN-FORCE type of gradient (Williams, 1992) estimator, which usually exhibits high variance and needs baselines to reduce it (Mohamed et al., 2020). This is in contrast to partitioned networks, which use standard, low-variance backpropagation for the hyperparameters and no separate validation set per client. To optimize the other hyperparameters, Khodak et al. (2021) wrapped FedEx with a traditional hyperparameter optimization strategy, the successive halving algorithm. This is orthogonal to our method and could be applied to partitioned networks as well. In Zhou et al. (2021), the authors perform a hyperparameter search independently on each client with some off-the-shelf methods and then aggregate the results of the search at the server once in order to identify the best hyperparameter setting. The main drawback of this method compared to partitioned networks is that when the local client datasets are small, a client-specific validation set is not informative, and the aggregation happens only once. Finally, there is also the recent work from Seng et al. (2022) which performs hyperparameter optimization and neural architecture search in the federated setting. Similarly to prior works, it requires client-specific validation data in order to optimize the hyperparameters.

## 5 EXPERIMENTS

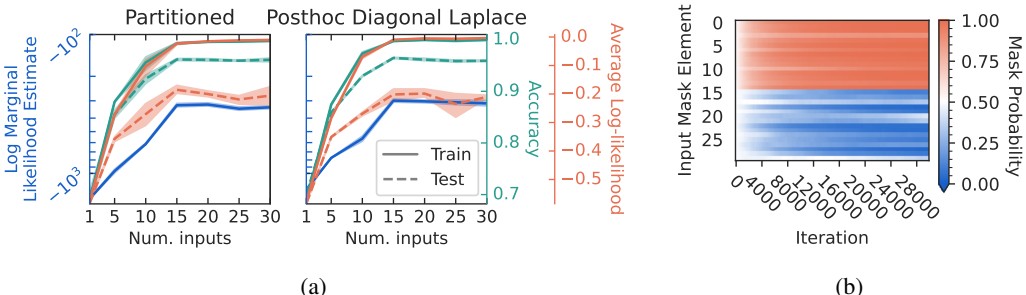

(a)                                       (b)

Figure 2: (a) Demonstrating the ability of the marginal-likelihood inspired objective $\mathcal{L}_{\mathrm{ML}}$ to identify the correct model on a toy input selection task. We plot the hyperparameter objective, train— and test--- set accuracy, and train— and test--- set log-likelihood with the partitioned networks method (left), and the post-hoc diagonal Laplace method (Immer et al., 2021) (right). (b) Mask over input features learned by partitioned networks over time. The first 15 features are correctly identified.

**Input Selection** To demonstrate that $\mathcal{L}_{\mathrm{ML}}$ is a good objective for model selection that captures the desirable properties of the marginal likelihood, we first deploy our method on the toy model selection task of Lyle et al. (2020): there the first 15 features are informative, and the remaining 15 are spurious

$$y \sim \mathrm{Bern}\left(\frac{1}{2}\right) \qquad \boldsymbol{x} = \big[\underbrace{y + \epsilon_1, \ldots, y + \epsilon_{15}}_{\text{Informative}}, \underbrace{\epsilon_{16}, \ldots, \epsilon_{30}}_{\text{Spurious}}\big]^{\mathsf{T}} \qquad \epsilon_1, \ldots, \epsilon_{30} \overset{\text{iid}}{\sim} \mathcal{N}(0, 1).$$

We specify a fixed mask over the inputs prior to training, where the first $K$ inputs remain unmasked, and the remainder is masked. We expect that, given multiple models with different (fixed) masks over the inputs, the proposed objective will be able to identify the correct one — *i.e.*, the one that keeps only the informative features. We train multiple fully connected neural networks (MLPs) on a training set of 1000 examples using our method and compare the final values of the $\mathcal{L}_{\mathrm{ML}}$ objective. The results are shown in Figure 2a. $\mathcal{L}_{\mathrm{ML}}$ correctly identifies 15 input features as the optimum, and correlates well with test accuracy and log-likelihood. Training loss and training accuracy, on the other hand, cannot alone disambiguate whether to use 15 or more input features.

**Differentiable input selection** We further show that we can learn the correct mask over the inputs in a differentiable manner using our method during a single training run. We parameterize a learnable mask over the inputs with a concrete Bernoulli distribution (Maddison et al., 2016) and treat the parameters of the mask distribution as a hyperparameter. We optimize them with respect to the proposed objective using our method. The evolution of the learned mask during training is shown in Figure 2b, where we see that we can correctly identify the first 15 informative features.

**Learning invariances through data-augmentations**    Following previous literature on learning soft invariances through learning data augmentations (Nabarro et al., 2022; van der Wilk et al., 2018; Benton et al., 2020; Schwöbel et al., 2021; Immer et al., 2022), we show that we can learn useful affine image augmentations, resulting in gains in test accuracy. We specify affine data augmentations as part of a probabilistic model as done by van der Wilk et al. (2018), averaging over multiple data augmentation samples during training and inference. This allows us to treat the data-augmentation distribution as a model hyperparameter rather than a training hyperparameter. For datasets, we consider MNIST, CIFAR10, TinyImagenet along with rotCIFAR10 and rotTinyImagenet, variants where the datapoints are randomly rotated at the beginning of training by angles sampled uniformly from $[-\pi, \pi]$ (Immer et al., 2022). Experimental setup details are provided in Appendix I.

For the CIFAR10 and rotCIFAR10 datasets, we consider as baselines standard training with no augmentations, Augerino (Benton et al., 2020) and Differentiable Laplace (Immer et al., 2022). Following Immer et al. (2022), we use $^{fix}_{up}$ResNets (Zhang et al., 2019) for the architectures. The results can be seen in Table 1. There, we observe that partitioned networks outperform all baselines in the case of CIFAR10 for both ResNet variants we consider. On RotCIFAR10, we observe that partitioned networks outperform the baseline and Augerino, but it is slightly outperformed by Differentiable Laplace, which optimizes additional prior hyperparameters. To demonstrate the scalability of partitioned networks, for the (rot)TinyImagenet experiments we consider a ResNet-50 architecture with GroupNorm(2). In Table 1 we observe that in both cases, partitioned networks learn invariances successfully and improve upon the baseline. Relative to Augerino, we observe that partitioned networks either improve (TinyImagenet) or are similar (rotTinyImagenet).

Table 1: Test accuracy with learning affine augmentations on (rot)CIFAR10 and (rot)TinyImagenet.

| Dataset | Architecture | Baseline | Augerino | Method Diff. Laplace | Partitioned |
|---|---|---|---|---|---|
| RotCIFAR10 | $^{fix}_{up}$ResNet-8 | $54.2_{\pm 0.4}$ | $75.4_{\pm 0.2}$ | $\mathbf{79.5_{\pm 0.6}}$ | $\mathbf{79.1_{\pm 0.0}}$ |
| CIFAR10 | $^{fix}_{up}$ResNet-8 | $74.1_{\pm 0.5}$ | $79.0_{\pm 1.0}$ | $84.2_{\pm 0.8}$ | $\mathbf{86.1_{\pm 0.4}}$ |
|  | $^{fix}_{up}$ResNet-14 | $79.5_{\pm 0.3}$ | $83.0_{\pm 0.1}$ | $88.1_{\pm 0.2}$ | $\mathbf{89.1_{\pm 0.8}}$ |
| RotTinyImagenet | ResNet-50 | $31.5_{\pm 0.6}$ | $\mathbf{44.5_{\pm 0.2}}$ | OOM[7] | $43.9_{\pm 0.3}$ |
| TinyImagenet | ResNet-50 | $44.2_{\pm 0.5}$ | $41.1_{\pm 0.2}$ | OOM | $\mathbf{48.6_{\pm 0.0}}$ |

Imbuing a model with useful invariances is particularly useful in the low-data regime, due to better data efficiency. To show that, we perform experiments where we artificially reduce the size of the training dataset. The results can be seen in Figure 3. We see that by learning augmentations with partitioned networks, we can drastically improve performance in the low-data regime upon a baseline that does not learn augmentations, while performing favorably against prior works in most cases.

On MNIST, our method outperforms the last-layer marginal-likelihood method (last-layer ML) by Schwöbel et al. (2021) in the large data regime but underperforms in the low-data regime. That is likely to be expected, as their work fits a Gaussian Process (GP) at the last layer (Wilson et al., 2016), which is better tailored for the low-data regime and results into a more flexible model (due to the GP corresponding to an additional, infinite width, layer). Since the MNIST-CNN is sufficiently small to fit multiple networks into memory, we also compare to a variant of our method where, instead of partitioning a single network, we train $C$ different networks where network $k$ is fit on data $\mathcal{D}_{1:k}$. This serves as an upper bound on the performance of the partitioned networks. We see that by partitioning a single network, we can achieve almost equivalent accuracy. On CIFAR10, partitioned networks outperform all other works on all data sizes we considered. On RotCIFAR10, partitioned networks perform again favourably, but they are marginally outperformed by differentiable Laplace in the low-data regime. Compared to partitioned networks where we only optimize augmentations, differentiable Laplace also optimizes the precision of a Gaussian prior over the weights, which better combats overfitting in the low-data regime. On both the TinyImagenet and rotTinyImagenet experiments we observe that partitioned networks either outperform or are similar to the baselines on all data sizes considered.

---

[7]Out of memory error on a 32GB Nvidia V100.

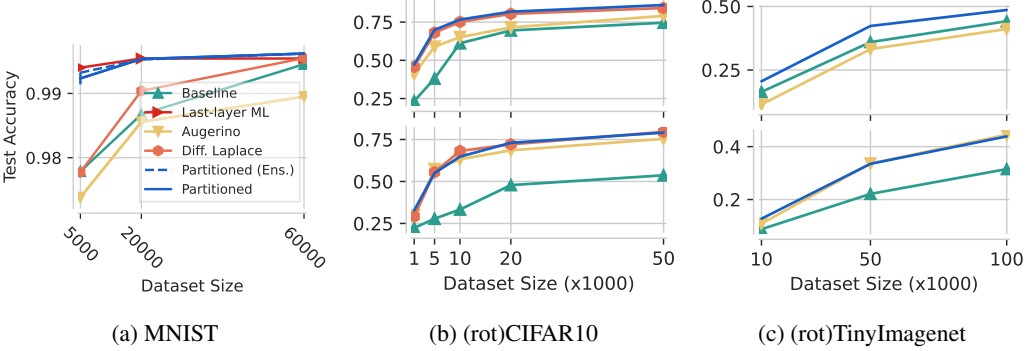

|              |              |                   |
|:------------:|:------------:|:-----------------:|
| (a) MNIST    | (b) (rot)CIFAR10 | (c) (rot)TinyImagenet |

Figure 3: Learning affine data augmentations on subsets of data. (b) uses a $^{\text{fix}}_{\text{up}}$ResNet-8 architecture whereas (c) a ResNet-50 architecture. (b,c) Top: normal dataset, bottom: rotated dataset.

**Comparisons to traditional training / validation split**   We further perform comparisons between partitioned networks and the more traditional training/validation split (denoted as validation set optimization) with additional finetuning to the task of learning data augmentations. This is realized as follows; we partition $20k$ CIFAR10 examples into training and validation data of specific proportions. We then either train a partitioned network (along with the hyperparameters on $\mathcal{L}_{\text{ML}}$) on these two chunks of data or train a standard network on the training set while using the validation set loss to obtain gradients for the data augmentation hyperparameters. For the validation set optimization baseline, once the hyperparameters are optimized, the resulting network is finetuned on the whole dataset for 20 epochs. The results for varying chunk proportions are provided in Table 2.

Table 2: Learning affine augmentations with $^{\text{fix}}_{\text{up}}$ResNet-14 on subset of CIFAR-10 ($20k$ examples). NaN denotes that a run crashed.

|                      | Chunk Proportions | | | | |
|----------------------|:----------:|:----------:|:----------:|:----------:|:----------:|
| Method               | $[0.3, 0.7]$ | $[0.5, 0.5]$ | $[0.7, 0.3]$ | $[0.8, 0.2]$ | $[0.9, 0.1]$ |
| Partitioned          | $\mathbf{82.9\%_{\pm 0.3}}$ | $\mathbf{83.0\%_{\pm 0.01}}$ | $\mathbf{83.7\%_{\pm 0.2}}$ | $\mathbf{84.0\%_{\pm 0.6}}$ | $\mathbf{84.6\%_{\pm 0.05}}$ |
| Validation set optim. | NaN | $78.9\%_{\pm 0.04}$ | $81.5\%_{\pm 0.2}$ | $82.6\%_{\pm 0.1}$ | $83.4\%_{\pm 0.1}$ |
| $\llcorner$ +Finetune | NaN | $81.3\%_{\pm 0.09}$ | $82.5\%_{\pm 0.2}$ | $83.5\%_{\pm 0.1}$ | $83.8\%_{\pm 0.3}$ |

We can see that partitioned networks (that do not employ additional finetuning) outperform validation set optimization with finetuning in all settings we tried. The gap does get smaller when we move to the more traditional 90/10 splits for train-

Table 3: Learning a feature extractor (first 2 out of 3 stages of a Wide ResNet-20) as a hyperparameter on CIFAR10.

| Method                | Chunk Proportions | Test accuracy |
|-----------------------|:-----------------:|:-------------:|
| Validation set optim. | $[0.9, 0.1]$      | $59.6\%_{\pm 0.6}$ |
| Partitioned           | $[0.1, 0.8, 0.1]$ | $\mathbf{87.3\%_{\pm 0.8}}$ |

ing/validation: a $10\%$ proportion for validation data is enough to optimize a handful of hyperparameters (just 6 scalars). To corroborate this claim, we set up an additional experiment; we use a Wide ResNet-20 on the full CIFAR10 dataset, where the first two out of the three stages (13 convolution layers) are considered as hyperparameters. The results for this setting can be seen in Table 3. We see that $10\%$ validation data are not enough, and the validation set optimization baseline performs poorly. This is in contrast to partitioned networks, where with three chunks, we can learn all of these hyperparameters successfully. Note that, compared to Augerino, applying partitioned networks to this setting is straightforward. To apply Augerino, one would have to come up with a metric that can be used to regularize the feature extractor towards "higher invariance".

**Partitioned networks for federated learning**   We consider federated learning (FL) (McMahan et al., 2017), a setting where data is distributed across many clients. In this setting, there are system properties that make hyperparameter optimization especially challenging (Wang et al., 2021). More specifically, obtaining a validation set and performing multiple training runs with different

hyperparameter settings might not be possible due to the additional communication and computation costs, and transient client availability (clients join and leave the training process at any time). Optimizing hyperparameters together with the model parameters in a single run is therefore especially beneficial (Wang et al., 2021), and partitioned networks are a good fit for FL.

We extend our centralized experimental setup to FL by splitting all $N$ clients into $C$ non-overlapping chunks, such that each chunk is understood as the union of all clients' data shards that belong to that chunk. During federated training, a client belonging to chunk $k$ sequentially optimizes partitions $\boldsymbol{w}_{k:C}$ through sub-networks $\boldsymbol{w}_s^{(k:C)}$ and computes a gradient wrt. the hyperparameters $\psi$. Note that partitions $\boldsymbol{w}_{1:k}$ remain unchanged and do not need to be communicated back to the server. This reduction in upload costs is a welcome property for FL, where upload costs can bottleneck system design. The server receives the (hyper-) parameter updates, averages them, and applies the result as a "gradient" to the server-side model in the traditional federated manner (Reddi et al., 2020). For partitioned networks, the hyperparameters that we optimize are the data augmentation parameters and, since we also include dropout in these architectures, the dropout rates (with the concrete relaxation from Maddison et al. (2016)). As a baseline, we consider the standard federated training without learning hyperparameters (denoted as FedAvg) as well as learning the augmentation parameters with Augerino Benton et al. (2020). Please see Appendix J for a detailed explanation of our FL setup.

Table 4 summarizes our results using different sub-sets and variations of MNIST and CIFAR10, where we also included rotMNIST Larochelle et al. (2007) as another dataset. We can see that partitioned networks allow training models that generalize better than both FedAvg and FedAvg with Augerino, at reduced communication costs. Especially when the true data-generating process and underlying source of non-i.i.d.-ness are explicitly accounted for — here in the form of rotation — the benefits of learning the augmentations with partitioned networks become apparent. For example, we observe that on the rotated datasets, partitioned networks learn to correctly increase the rotation angle.

Table 4: Validation accuracy averaged over the last 10 evaluations, each 10 rounds apart; standard-error is computed across 4 random seeds. All datasets are adapted to the federated setting and are synthetically split to be non-i.i.d. sampled as described in Appendix J.2.

| Dataset & size | ↑MNIST | | | ↑RotMNIST | | | ↓Upload |
| Method | 1.25k | 5k | 50k | 1.25k | 5k | 50k | [%] |
|---|---|---|---|---|---|---|---|
| FedAvg | $95.4\%_{\pm 0.1}$ | $97.4\%_{\pm 0.1}$ | $99.0\%_{\pm 0.1}$ | $80.5\%_{\pm 0.0}$ | $90.4\%_{\pm 0.5}$ | $96.8\%_{\pm 0.1}$ | 100 |
| FedAvg + Augerino | $94.2\%_{\pm 0.5}$ | $96.4\%_{\pm 0.1}$ | $99.1\%_{\pm 0.0}$ | $79.5\%_{\pm 0.3}$ | $89.0\%_{\pm 2.0}$ | $95.3\%_{\pm 0.2}$ | 100 |
| FedAvg + Partitioned | $\mathbf{97.0\%_{\pm 0.1}}$ | $\mathbf{98.3\%_{\pm 0.0}}$ | $99.2\%_{\pm 0.1}$ | $\mathbf{85.7\%_{\pm 0.9}}$ | $\mathbf{93.5\%_{\pm 0.6}}$ | $\mathbf{97.8\%_{\pm 0.1}}$ | 77 |

| | ↑CIFAR10 | | | ↑RotCIFAR10 | | | ↓Upload |
| | 1.25k | 5k | 45k | 1.25k | 5k | 45k | [%] |
|---|---|---|---|---|---|---|---|
| FedAvg | $50.2\%_{\pm 0.4}$ | $64.5\%_{\pm 0.3}$ | $79.2\%_{\pm 0.7}$ | $35.6\%_{\pm 0.3}$ | $45.2\%_{\pm 0.1}$ | $53.9\%_{\pm 1.1}$ | 100 |
| FedAvg + Augerino | $49.9\%_{\pm 0.8}$ | $65.0\%_{\pm 0.2}$ | $79.9\%_{\pm 0.4}$ | $36.1\%_{\pm 0.2}$ | $45.0\%_{\pm 0.2}$ | $56.4\%_{\pm 0.7}$ | 100 |
| FedAvg + Partitioned | $50.8\%_{\pm 1.0}$ | $64.8\%_{\pm 0.4}$ | $\mathbf{81.5\%_{\pm 0.5}}$ | $\mathbf{37.1\%_{\pm 0.2}}$ | $45.3\%_{\pm 0.3}$ | $\mathbf{60.6\%_{\pm 0.2}}$ | 91 |

## 6 DISCUSSION

We propose partitioned networks as a new method for hyperparameter optimization inspired by the marginal likelihood objective. It provides a general and scalable solution to finding hyperparameters in a single training run without requiring access to a validation set while introducing less additional overhead to the training task than existing approaches. We showed that partitioned networks are applicable on a wide range of tasks; they can identify the correct model on illustrative toy examples, they can learn data augmentations in a way that improves data efficiency, they can optimize general feature extractors as hyperparameters and they can also optimize dropout rates. In the federated setting, partitioned networks allow us to overcome practical challenges, reduce the communication overhead and obtain better models. The notion of partitioned networks we propose in this work is novel to the literature and an orthogonal approach to many existing hyperparameter tuning algorithms. Like any other method, partitioned networks come with their own limitations, e.g., needing a partitioning strategy. We expand upon them in appendix H. We hope to see our method successfully reducing the need to perform hyperparameter search through repeated training and thereby contribute to the community's effort to reduce its carbon footprint.

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

## A $\mathcal{L}_{\mathrm{ML}}$ IS A LOWER-BOUND TO THE MARGINAL LIKELIHOOD

In this section, we show that the objective in equation 2 is a lower-bound on the marginal likelihood, under a mild assumption on each approximate posterior $q_k(\boldsymbol{w})$. The aim is to approximate:

$$\log p(\mathcal{D}|\psi) = \sum_{k=1}^{C} \log p(\mathcal{D}_k|\mathcal{D}_{1:k-1}, \psi) \tag{5}$$

Our partitioned approximation is given by:

$$\sum_{k=1}^{C} \mathbb{E}_{q_{k-1}(\boldsymbol{w})} \left[ \log p(\mathcal{D}_k|\boldsymbol{w}, \psi) \right] \tag{6}$$

We can get the equation for the gap between quantities in 5 and 6:

$$\mathrm{gap} = \sum_{k=1}^{C} \log p(\mathcal{D}_k|\mathcal{D}_{1:k-1}, \psi) - \sum_{k=1}^{C} \mathbb{E}_{q_{k-1}(\boldsymbol{w})} \left[ \log p(\mathcal{D}_k|\boldsymbol{w}, \psi) \right] \tag{7}$$

$$= \sum_{k=1}^{C} \mathbb{E}_{q_{k-1}(\boldsymbol{w})} \left[ \log p(\mathcal{D}_k|\mathcal{D}_{1:k-1}, \psi) - \log p(\mathcal{D}_k|\boldsymbol{w}, \psi) \right] \tag{8}$$

$$= \sum_{k=1}^{C} \mathbb{E}_{q_{k-1}(\boldsymbol{w})} \left[ \log \frac{p(\mathcal{D}_k|\mathcal{D}_{1:k-1}, \psi)}{p(\mathcal{D}_k|\boldsymbol{w}, \psi)} \right] \tag{9}$$

$$= \sum_{k=1}^{C} \mathbb{E}_{q_{k-1}(\boldsymbol{w})} \left[ \log \frac{\overbrace{p(\boldsymbol{w}|\mathcal{D}_{1:k}, \psi) p(\mathcal{D}_k|\mathcal{D}_{1:k-1}, \psi) p(\boldsymbol{w}|\mathcal{D}_{1:k-1}, \psi)}^{p(\boldsymbol{w}, \mathcal{D}_k|\mathcal{D}_{1:k-1})}}{p(\boldsymbol{w}|\mathcal{D}_{1:k}, \psi) \underbrace{p(\mathcal{D}_k|\boldsymbol{w}, \psi) p(\boldsymbol{w}|\mathcal{D}_{1:k-1}, \psi)}_{p(\boldsymbol{w}, \mathcal{D}_k|\mathcal{D}_{1:k-1})}} \right] \tag{10}$$

$$= \sum_{k=1}^{C} \mathbb{E}_{q_{k-1}(\boldsymbol{w})} \left[ \log \frac{p(\boldsymbol{w}|\mathcal{D}_{1:k-1}, \psi)}{p(\boldsymbol{w}|\mathcal{D}_{1:k}, \psi)} \right] \tag{11}$$

$$= \sum_{k=1}^{C} D_{\mathrm{KL}} \left[ q_{k-1}(\boldsymbol{w}) \| p(\boldsymbol{w}|\mathcal{D}_{1:k}, \psi) \right] - D_{\mathrm{KL}} \left[ q_{k-1}(\boldsymbol{w}) \| p(\boldsymbol{w}|\mathcal{D}_{1:k-1}, \psi) \right] \tag{12}$$

We now make two assumptions

- $D_{\mathrm{KL}} \left[ q_{k-1}(\boldsymbol{w}) \| p(\boldsymbol{w}|\mathcal{D}_{1:k}, \psi) \right] \geq D_{\mathrm{KL}} \left[ q_k(\boldsymbol{w}) \| p(\boldsymbol{w}|\mathcal{D}_{1:k}, \psi) \right]$. This is motivated from the fact that $q_k(\boldsymbol{w})$ is trained on all data chunks $\mathcal{D}_{1:k}$ so it is expected to be a better approximation to the posterior $p(\boldsymbol{w}|\mathcal{D}_{1:k})$, compared to $q_{k-1}(\boldsymbol{w})$ which is only trained on $\mathcal{D}_{1:k-1}$.

- $D_{\mathrm{KL}} \left[ q_{C-1}(\boldsymbol{w}) \| p(\boldsymbol{w}|\mathcal{D}_{1:C}, \psi) \right] \geq D_{\mathrm{KL}} \left[ q_0(\boldsymbol{w}) \| p(\boldsymbol{w}) \right]$. Since we are free to choose the approximate posterior before seeing any data — $q_0(\boldsymbol{w})$—, we can set it to be equal to the prior $p(\boldsymbol{w})$ which, together with the positivity of the KL divergence, trivially satisfies this assumption.

Therefore, by rearranging Eq. 12 and using our two assumptions we have that the gap is positive

$$\mathrm{gap} = -D_{\mathrm{KL}} \left[ q_0(\boldsymbol{w}) \| p(\boldsymbol{w}) \right] + D_{\mathrm{KL}} \left[ q_{C-1}(\boldsymbol{w}) \| p(\boldsymbol{w}|\mathcal{D}_{1:C}, \psi) \right] +$$
$$\sum_{k=1}^{C} D_{\mathrm{KL}} \left[ q_{k-1}(\boldsymbol{w}) \| p(\boldsymbol{w}|\mathcal{D}_{1:k}, \psi) \right] - D_{\mathrm{KL}} \left[ q_k(\boldsymbol{w}) \| p(\boldsymbol{w}|\mathcal{D}_{1:k}, \psi) \right] \geq 0, \tag{13}$$

and our approximation is a lower bound to the marginal likelihood, *i.e.*,

$$\log p(\mathcal{D}|\psi) \geq \sum_{k=1}^{C} \mathbb{E}_{q_{k-1}(\boldsymbol{w})} \left[ \log p(\mathcal{D}_k|\boldsymbol{w}, \psi) \right]. \tag{14}$$

## B PARTITIONED NETWORKS AS A SPECIFIC APPROXIMATION TO THE MARGINAL LIKELIHOOD

In this section of the appendix, we show that the partitioned neural networks we presented in the paper are a particular instance of the approximation to the marginal likelihood shown in equation 2.

Consider a dataset $\mathcal{D}$ comprised of $C$ shards, i.e. $\mathcal{D} = (\mathcal{D}_1, \ldots, \mathcal{D}_C)$, along with a model, e.g., a neural network, with parameters $\boldsymbol{w} \in \mathbb{R}^{D_w}$, a prior $p(\boldsymbol{w}) = \prod_{j=1}^{D_w} \mathcal{N}(\boldsymbol{w}_j | 0, \lambda)$ and a likelihood $p(\mathcal{D}|\boldsymbol{w}, \psi)$ with hyperparameters $\psi$. Assuming a sequence over the dataset chunks, we can write out the true marginal likelihood as

$$\log p(\mathcal{D}|\psi) = \sum_k \log p(\mathcal{D}_k | \mathcal{D}_{1:k-1}, \psi) = \sum_k \log \mathbb{E}_{p(\boldsymbol{w}|\mathcal{D}_{1:k-1}, \psi)} \left[ p(\mathcal{D}_k|\boldsymbol{w}, \psi) \right] \tag{15}$$

$$\geq \sum_k \mathbb{E}_{p(\boldsymbol{w}|\mathcal{D}_{1:k-1}, \psi)} \left[ \log p(\mathcal{D}_k|\boldsymbol{w}, \psi) \right]. \tag{16}$$

Since the true posteriors $p(\boldsymbol{w}|\mathcal{D}_{1:j}, \psi)$ for $j \in \{1, \ldots, C\}$ are intractable, we can use variational inference to approximate them with $q_{\phi_j}(\boldsymbol{w})$ for $j \in \{1, \ldots, C\}$, with $\phi_j$ being the to-be-optimized parameters of the $j$'th variational approximation. Based on the result from Appendix A, when $q_{\phi_j}(\boldsymbol{w})$ are optimized to match the respective posteriors $p(\boldsymbol{w}|\mathcal{D}_{1:j}, \psi)$, we can use them to approximate the marginal likelihood as

$$\log p(\mathcal{D}|\psi) \geq \sum_k \mathbb{E}_{q_{\phi_{k-1}}(\boldsymbol{w})} \left[ \log p(\mathcal{D}_k|\boldsymbol{w}, \psi) \right]. \tag{17}$$

Partitioned networks correspond to a specific choice for the sequence of approximating distribution families $q_{\phi_k}(\boldsymbol{w})$. Specifically, we partition the parameter space $\boldsymbol{w}$ into $C$ chunks, i.e., $\boldsymbol{w}_k \in \mathbb{R}^{D_{wk}}$, such that $\sum_k D_{wk} = D_w$, and we associate each parameter chunk $\boldsymbol{w}_k$ with a data shard $\mathcal{D}_k$. Let $r_{\phi_k}(\boldsymbol{w}_k)$ be base variational approximations over $\boldsymbol{w}_k$ with parameters $\phi_k$. Each approximate distribution $q_{\phi_k}(\boldsymbol{w})$ is then defined in terms of these base approximations, i.e.,

$$q_{\phi_k}(\boldsymbol{w}) = \left( \prod_{j=1}^{k-1} r_{\phi_j}(\boldsymbol{w}_j) \right) r_{\phi_k}(\boldsymbol{w}_k) \left( \prod_{m=k+1}^{K} r_0(\boldsymbol{w}_m) \right) \tag{18}$$

where $r_0(\cdot)$ is some base distribution with no free parameters. In accordance with the assumptions in appendix A, we can then fit each $q_{\phi_k}(\boldsymbol{w})$ by minimising the KL-divergence to $p(\boldsymbol{w}|\mathcal{D}_{1:k}, \psi)$ – the posterior after seeing $k$ chunks:

$$D_{\mathrm{KL}} \left[ q_{\phi_k}(\boldsymbol{w}) \| p(\boldsymbol{w}|\mathcal{D}_{1:k}, \psi) \right] = - \mathbb{E}_{q_{\phi_k}(\boldsymbol{w})} [\log p(\mathcal{D}_{1:k}|\boldsymbol{w}, \psi)] + D_{\mathrm{KL}} \left[ q_{\phi_k}(\boldsymbol{w}) \| p(\boldsymbol{w}) \right]$$
$$+ \log p(\mathcal{D}_{1:k}|\psi) \tag{19}$$
$$\tag{20}$$

Finding the optimum with respect to $\phi_k$:

$$\underset{\phi_k}{\arg\min} \ D_{\mathrm{KL}} \left[ q_{\phi_k}(\boldsymbol{w}) \| p(\boldsymbol{w}|\mathcal{D}_{1:k}, \psi) \right] = \tag{21}$$

$$= \underset{\phi_k}{\arg\min} - \mathbb{E}_{q_{\phi_k}(\boldsymbol{w})} [\log p(\mathcal{D}_{1:k}|\boldsymbol{w}, \psi)] + D_{\mathrm{KL}} \left[ q_{\phi_k}(\boldsymbol{w}) \| p(\boldsymbol{w}) \right] \tag{22}$$

$$= \underset{\phi_k}{\arg\min} - \mathbb{E}_{q_{\phi_k}(\boldsymbol{w})} [\log p(\mathcal{D}_{1:k}|\boldsymbol{w}, \psi)]$$

$$+ D_{\mathrm{KL}} \left[ \left( \prod_{j=1}^{k-1} r_{\phi_j}(\boldsymbol{w}_j) \right) r_{\phi_k}(\boldsymbol{w}_k) \left( \prod_{m=k+1}^{K} r_0(\boldsymbol{w}_m) \right) \| \prod_{i}^{K} p(\boldsymbol{w}_i) \right] \tag{23}$$

$$= \underset{\phi_k}{\arg\min} - \mathbb{E}_{q_{\phi_k}(\boldsymbol{w})} [\log p(\mathcal{D}_{1:k}|\boldsymbol{w}, \psi)] + D_{\mathrm{KL}} \left[ r_{\phi_k}(\boldsymbol{w}_k) \| p(\boldsymbol{w}_k) \right]. \tag{24}$$

We can now obtain partitioned networks by assuming that $r_{\phi_k}(\boldsymbol{w}_k) = \mathcal{N}(\boldsymbol{w}_k | \phi_k, \nu \mathbf{I})$ for $k \in \{1, \ldots, C\}$, $r_0(\mathbf{w}) = \mathcal{N}(\boldsymbol{w} | \hat{\boldsymbol{w}}, \nu \mathbf{I})$, with $\hat{\boldsymbol{w}}$ being the parameters at initialization (i.e., before we

update them on data) and taking $\nu \to 0$, i.e., in machine-precision, the weights are deterministic. As noted in Section I.1, we scale the weight-decay regularizer for $\phi_k$ (whenever used) differently for each partition $k$, such that it can be interpreted as regularization towards a prior. In the experiments where we do not regularize $\phi_k$ according to $p(\mathbf{w}_k)$ when we optimize them, this implicitly corresponds to $\lambda \to \infty$ (i.e. the limiting behaviour when the variance of $p(\mathbf{w})$ goes to infinity), which makes the contribution of the regularizer negligible.

## C  PARTITIONING SCHEMES

There are several ways in which we could aim to partition the weights of a neural network. Throughout the experimental section 5, we partition the weights by assigning a fixed proportion of weights in each layer to a given partition at random. We call this approach **random weight partitioning**.

We also experimented with other partitioning schemes. For example, we tried assigning a fixed proportion of a layer's outputs (*e.g.*, channels in a convolution layer) to each partition. All weights in a given layer that a specific output depends on would then be assigned to that partition. We call this approach **node partitioning**. Both approaches are illustrated in Figure 4.

One benefit of the node partitioning scheme is that it makes it possible to update multiple partitions with a single batch; This is because we can make a forward pass at each linear or convolutional layer with the full network parameters $\boldsymbol{w}$, and, instead, mask the appropriate inputs and outputs to the layer to retrieve an equivalent computation to that with $\boldsymbol{w}_s^{(k)}$. The gradients also need to be masked on the backward pass adequately. No such simplification is possible with the random weight partitioning scheme; if we were to compute a backward pass for a single batch of examples using different subnetworks for each example, the memory overhead would grow linearly with the number of subnetworks used.

In initial experiments, we found both *random weight partitioning* and *node partitioning* performed similarly. In the experimental section 5, we focused on the former, as it's easier to reason about with relation to *e.g.*, dropout.

Throughout this work, partitioning happens prior to initiating training, and remains fixed throughout. It might also be possible to partition the network parameters dynamically during training, which we leave for future work.

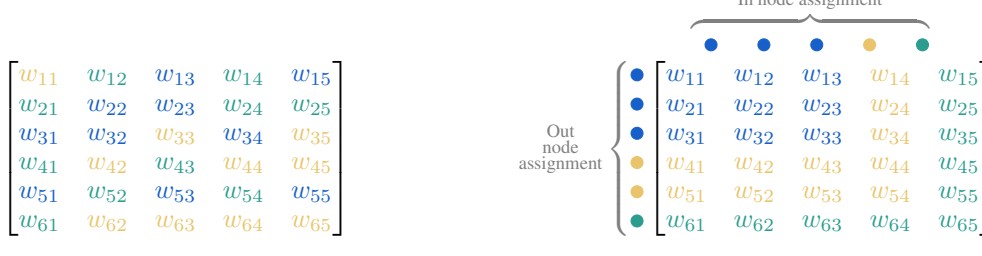

(a) Random weight partitioned          (b) Node partitioned

Figure 4: Figures showing how the weights within a single weight matrix $W \in \mathbb{R}^{6 \times 5}$ for a linear layer would be partitioned.

## D  SCALABILITY

In the paper, we claim that our method is scalable compared to Schwöbel et al. (2021) and Immer et al. (2022). What constraints the scalability of the mentioned prior works, however, is different.

For the Last Layer Marginal Likelihood, although the approach works on small datasets such as PCAM (Veeling et al., 2018) and MNIST, the authors report that they were unable to learn invariances

on larger datasets such as CIFAR10. In (Schwöbel et al., 2021, section 7), they explore the issue of scalability in more detail, and showcase that last layer marginal likelihood is insufficient.

Differentiable Laplace performs well, even on more complex datasets, such as CIFAR10. Their scalability, however, is limited by the computational and memory complexity of their method, which we go into in more detail in the section below.

## D.1 Complexity Analysis

First, we consider the scalability of our algorithm in terms of computational and memory complexity. In particular, we show that our method scales much more favourably compared to Differentiable Laplace (Immer et al., 2022).

We present our analysis for a feed-forward model of depth $L$, with layer widths $D$[8]. In order to directly compare to Immer et al. (2022) and Benton et al. (2020), we consider the complexities in the invariance learning setup (Benton et al., 2020; van der Wilk et al., 2018) with $S$ augmentation samples. In other experiments, hyperparameter optimization setups, $S$ can be taken to be 1. The notation is summarized in Table 5.

| | |
|---|---|
| $N$ | Number of datapoints in dataset $\mathcal{D}$ |
| $N_B$ | Batch size |
| $S$ | Number of augmentation samples[9] |
| $C$ | Output size (number of classes) |
| $D$ | Feedforward network layer widths |
| $L$ | Feedforward network depth |
| $P$ | Number of parameters (s.t. $\mathcal{O}(P) = \mathcal{O}(LD^2 + DC)$) |

Table 5: Notation for complexity analysis.

We consider the computational and memory costs of 1) obtaining a gradient with respect to the parameters 2) obtaining a gradient with respect to the hyperparameters, and 3) computing the value of the model/hyperparameter selection objective for each method. All analysis assumes computation on a Monte-Carlo estimate of the objective on a single batch of data.

In Tables 6 and 7, we assume that $C < D$, and hence, for the clarity of comparison, sometimes fold a factor depending $C$ into a factor depending on $D$ if it's clearly smaller. This hiding of the factors was only done for Differentiable Laplace, which is the worst scaling method.

### D.1.1 Computational Complexity

| | Param. Backward | Hyperparam. Backward | Hyperparam. Objective |
|---|---|---|---|
| Partitioned | $\mathcal{O}(N_B PS)$ | $\mathcal{O}(N_B PS)$ | $\mathcal{O}(N_B PS)$ |
| Augerino | $\mathcal{O}(N_B PS)$ | $\mathcal{O}(N_B PS)$ | $\mathcal{O}(N_B PS)$ |
| Diff. Laplace | $\mathcal{O}(N_B PS)$ | $\mathcal{O}(N_B PS + NCP + NCDLS + LD^3)$ | $\mathcal{O}(NPS + NCP + NCDLS + LD^3)$ |

Table 6: **Computational Complexities**. The two terms highlighted for Augerino can be computed in a single backward pass. For Differentiable Laplace, the terms in blue can be amortised over multiple hyperparameter backward passes. That is why, in their method, they propose updating the hyperparameters once every epoch on (possibly) multiple batches of data, rather than once on every batch as is done with Partitioned Networks and Augerino.

---

[8]This is for the ease of comparison. Same upper bound complexities will hold for a network of variable sizes $D_\ell$ for $\ell \in [L]$, where $D = \max_\ell D_\ell$

[9]Only relevant for invariance learning.

### D.1.2 MEMORY COMPLEXITY

The memory complexities for Partitioned Networks, Augerino, and Differentiable Laplace are shown in Table 7. Crucially, the memory required to update the hyperparameters for Differentiable Laplace scales as $\mathcal{O}(N_B SLD^2 + P)$, with a term depending on the square of the network widths. This can become prohibitively expensive for larger models, and is likely the reason why their paper only considers experiments on architectures with widths up to a maximum of 256.

|  | Param. Backward | Hyperparam. Backward | Hyperparam. Objective |
|---|---|---|---|
| Partitioned | $\mathcal{O}(N_B SLD + P)$ | $\mathcal{O}(N_B SLD + P)$ | $\mathcal{O}(N_B SD + P)$ |
| Augerino | $\mathcal{O}(N_B SLD + P)$ | $\mathcal{O}(N_B SLD + P)$ | $\mathcal{O}(N_B SD + P)$ |
| Diff. Laplace | $\mathcal{O}(N_B SLD + P)$ | $\mathcal{O}(N_B SLD^2 + P)$ | $\mathcal{O}(N_B SLD^2 + P)$ |

Table 7: **Memory Complexities**. Differences are highlighted in red.

### D.2 PRACTICAL SCALABILITY

A complexity analysis in big-$\mathcal{O}$ notation as provided by us in the previous sections allows to understand scalability in the limit, but constant terms that manifest in practice are still of interest. In this section we aim present real timing measurements for our method in comparison to Augerino and Differential Laplace, and elaborate on what overhead might be expected with respect to standard neural network training.

The empirical timings measurements on an NVIDIA RTX 3080-10GB GPU are shown in Table 8. We used a batch-size of 250, 200 for the MNIST and CIFAR10 experiments respectively, and 20 augmentation samples, just like in our main experiments in Table 1 and Figure 3. As can be seen, the overhead from using a partitioned network is fairly negligible compared to a standard forward and backward pass. The one difference compared to Augerino is, however, the fact that a separate forward-backward pass needs to be made to update the hyperparameters and regular parameters. This necessity is something that can be side-stepped with alternative partitioning schemes, as preliminarily mentioned in appendix C, and is an interesting direction for future research.

| Method | | MNIST CNN | CIFAR10 $^{fix}_{up}$ResNet-8 | $^{fix}_{up}$ResNet-14 |
|---|---|---|---|---|
| Augerino | | $\times 1$ | $\times 1$ | $\times 1$ |
| Diff. Laplace$^\dagger$ | Param. | $\times 1$ | $\times 1$ | $\times 1$ |
|  | Hyperparam. | $\times 2015.6$ | $\times 18.2$ | - |
| Partitioned | Param. | $\times 1.08$ | $\times 1.17$ | $\times 1.21$ |
|  | Hyperparam. | $\times 1.08$ | $\times 1.08$ | $\times 1.09$ |

Table 8: Relative empirical time increase with respect to a regular parameter update during standard training. $\dagger$ The timing multipliers with respect to the baseline for $^{fix}_{up}$ResNet-8 are taken from the timings reported in (Immer et al., 2022, Appendix D.4). On the ResNet-14, we get an out-of-memory error during the hyperparam. update step with Differentiable Laplace on the NVIDIA RTX 3080-10GB GPU when running with the official codebase (Immer and van der Ouderaa).

**Memory Overhead** Our proposed method's memory consumption scales in the same way as Augerino or vanilla neural network training. There is a minor constant memory overhead due to having to store the assignment of weights to partitions. In general, only $\log C$ bits per parameter are necessary to store the partition assignments, where $C$ is the number of chunks. In our implementation, we only consider $C < 2^8$, and hence store the assignments in byte tensors. This means that the partitioned models require extra $25\%$ memory for storing the parameters (when using 32bit floats to represent the parameters).

If the "default" weight values (i.e. those denoted $\hat{\boldsymbol{w}}_i$ in Figure 1) are non-zero, there is an additional overhead to storing those as well, which doubles the memory required to store the parameters. We observed there was no difference in performance when setting default weight values to $0$ in architectures in which normalisation layers are used (i.e. most modern architectures). As such, we would in general recommend to set the default weight values to $0$. However, we found setting default values to the initialised values to be necessary for stability of training deep normalisation-free architectures such as the $\mathrm{fix}_{\mathrm{up}}$ architectures (Zhang et al., 2019) we used to compare with Differentiable Laplace. As their method is not compatible with BatchNorm, we used these architectures in our experiments, and hence used non-zero default values.

Lastly, if the default weight values are set to the (random) initialisation values, it is possible to write a cleverer implementation in which only the random seeds are stored in memory, and the default values are re-generated every time they are need in a forward and a backward pass. This would make the memory overhead from storing the default values negligible.

## E    NOTE ON AUGERINO

In replicating Augerino (Benton et al., 2020) within our code-base and experimenting with the implementation, we discovered a pathological behaviour that is partly mirrored by the authors of Immer et al. (2022). In particular, note that the loss function (Benton et al., 2020, Equation (5)) proposed by the authors is problematic in the sense that for any regularization strength $\lambda > 0$, the optimal loss value is negative infinity since the regularization term (negative L2-norm) is unbounded. In our experiments we observe that for a sufficiently-large value of $\lambda$ and after a sufficient number of iterations, this behaviour indeed appears and training diverges. In practice, using Augerino therefore necessitates either careful tuning of $\lambda$, clipping the regularisation term (a method that introduces yet another hyperparameter), or other techniques such as early stopping.

In the open-source repository for the submission (Benton et al.), it can be seen that on many experiments the authors use a "safe" variant of the objective, in which they clip the regulariser (without pass-through of the gradient) once the $l_\infty$-norm of any of the hyperparameters becomes larger than an arbitrary threshold. Without using this adjustment, we found that the Augerino experiments on MNIST crashed every time with hyperparameters diverging to infinity.

## F    SENSITIVITY TO PARTITIONING

### F.1    SENSITIVITY IN TERMS OF FINAL PERFORMANCE

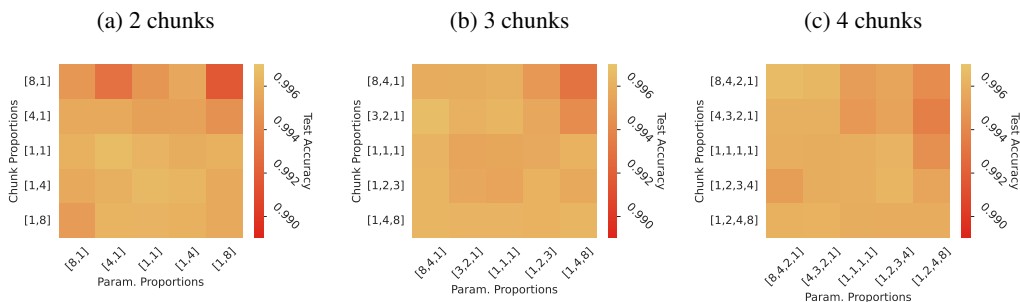

Figure 5: Learning affine augmentations on **MNIST** with a CNN fit on all data. $x$- and $y-$ ticks denote the ratios of parameters/datapoints assigned to each partition/chunk respectively.

Partitioned networks allow for learning hyperparameters in a single training run, however, they introduce an additional hyperparameter in doing so: the partitioning scheme. The practitioner needs to choose the number of chunks $C$, the relative proportions of data in each chunk, and the relative proportions of parameters assigned to each of the $C$ partitions $\boldsymbol{w}_k$. We investigate the sensitivity to the partitioning scheme here. We show that our results are fairly robust to partitioning through a grid-search over parameter partitions and chunk proportions on the affine augmentation learning task on MNIST with the CNN architecture we use throughout this work.

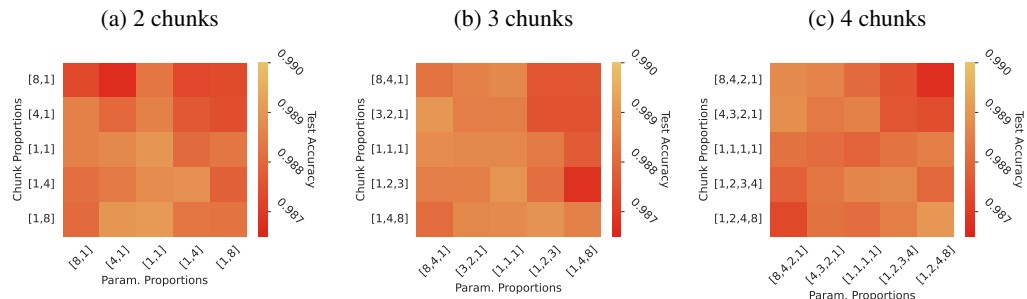

Figure 6: Learning affine augmentations on **RotMNIST** with a CNN fit on all data. $x$- and $y-$ ticks denote the ratios of parameters/datapoints assigned to each partition/chunk respectively.

Figure 5 and Figure 6 show the test accuracy for a choice of chunk and parameter proportions across two, three and four chunks. The proportions are to be read as un-normalized distributions; for example, chunk proportions set to $[1, 8]$ denotes that there are $8\times$ as many datapoints assigned to the second compared to the first. Each configuration was run with 2 random seeds, and we report the mean across those runs in the figure. The same architecture used was the same as for the main MNIST experiments in section 5 (see Appendix I.4 for details).

We observe that for various partition/dataset-chunking configurations, all models achieve fairly similar final test accuracy. There is a trend for models with a lot of parameters assigned to later chunks, but with few datapoints assigned to later chunks, to perform worse. While these results show a high level of robustness against the choice of additional hyperparameters introduced by our method, these results do show an opportunity or necessity for choosing the right partitioning scheme in order to achieve optimal performance.

## F.2 SENSITIVITY IN TERMS OF HYPERPARAMETERS FOUND

To compare how the different partitioning schemes qualitatively impact the hyperparameters that the method identifies, we also retrain vanilla models from scratch using the hyperparameter values found using partitioned networks. Namely, we take the final value of the hyperparameters learned with partitioned networks with a given partitioning scheme, and plot the final test set accuracy of a vanilla neural network model trained from scratch with those hyperparameters. The results are shown in Figures 7 and 8.

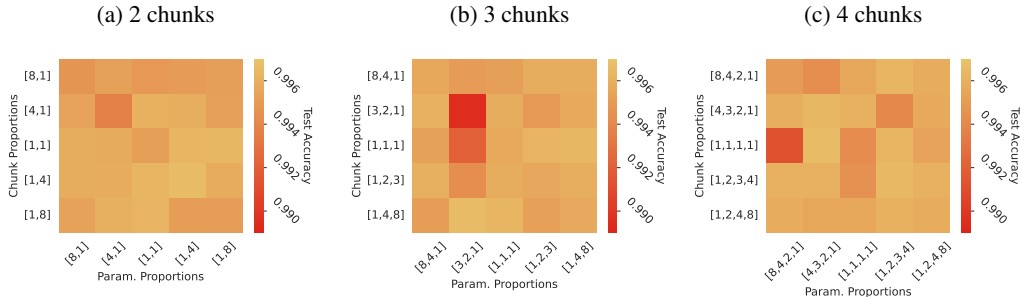

Figure 7: Standard neural network trained on **MNIST** with a CNN fit on all data, with hyperparameters found using partitioned networks with chunk and parameter proportions corresponding to those in Figure 5. $x$- and $y-$ ticks denote the ratios of parameters/datapoints assigned to each partition/chunk respectively.

## G HOW GOOD ARE THE HYPERPARAMETERS FOUND?

Here we show that the hyperparameters found by partitioned networks are also a good set of hyperparameters for vanilla neural networks retrained from scratch. This section expands on the

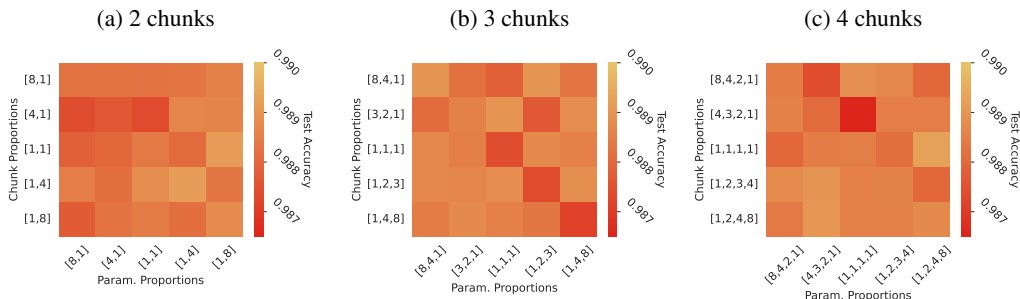

Figure 8: Standard neural network trained on **RotMNIST** with a CNN fit on all data, with hyperparameters found using partitioned networks with chunk and parameter proportions corresponding to those in Figure 6. $x$- and $y-$ ticks denote the ratios of parameters/datapoints assigned to each partition/chunk respectively.

experiment in section F.2. To validate this claim, we conducted a fairly extensive hyperparameter search on the affine augmentation learning task on RotMNIST; we trained 200 models by first sampling a set of affine augmentation parameters uniformly at random from a predefined range[10], and then training a neural network model (that averages across augmentation samples at train and test time, as described in Benton et al. (2020)) with standard neural training with those hyperparameters fixed throughout.

In Figure 9, we plot the final test-set performance of all the models trained with those hyperparameters sampled from a fixed range. Alongside, we show the hyperparameters and test-set performance of the partitioned networks as they progress throughout training. The partitioned networks consistently achieve final test-set performance as good as that of the best hyperparameter configurations identified through extensive random sampling of the space. We also show the test-set performance of neural network models, trained through standard training, with hyperparameters fixed to the final hyperparameter values identified by the partitioned networks. The hyperparameters identified by partitioned networks appear to also be good for regular neural networks; the standard neural networks with hyperparameters identified through partitioned training also outperform the extensive random sampling of the hyperparameter space. Furthermore, Figure 9 shows that partitioned networks do learn full rotation invariance on the RotMNIST task, i.e. when full rotation invariance is present in the data generating distribution.

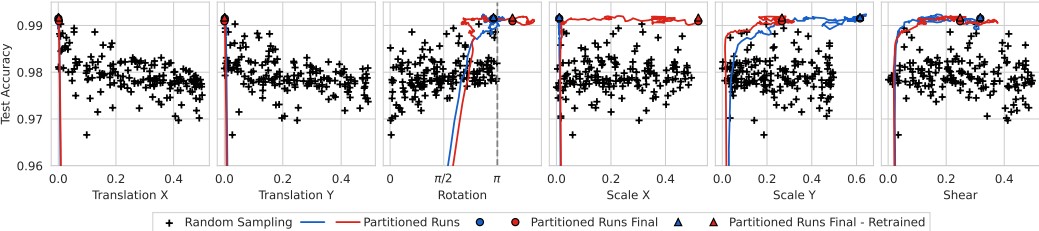

Figure 9: The test-set performance plotted alongside (1D projections of) affine augmentation hyperparameters on the RotMNIST task with MNIST-CNN. Final test-set accuracies are shown for the hyperparameters sampled randomly for a neural network model trained through standard training with those hyperparameters fixed (+). For multiple partitioned networks runs, the plot shows the progression of the identified hyperparameters and the test-set performance through the training run (———), as well as the final hyperparameters and test-set performance (● ●). Lastly, the plot also shows the final test-set accuracies of models trained through standard training on the final hyperparameters identified through partitioned training (▲ ▲).

---

[10]The ranges were: $\mathrm{Uniform}(0, \pi)$ for the maximum rotation, and $\mathrm{Uniform}(0, \frac{1}{2})$ for all the remaining affine augmentation parameters (maximum shear, maximum $x-$ and $y-$translation, and maximum $x-$ and $y-$ scale).

# H    LIMITATIONS

As mentioned in the main text, our method improves upon existing work, but also comes with its own limitations.

**Complexity**    Inherent to our method — as presented in e.g. Figure 1 — is the necessity for an additional forward-backward pass to update the hyperparameters. Consequently, hyperparameter optimization has additional costs which, however, are significantly less than the computational costs of existing work, as we discuss in more detail in Appendix D.1 and the experimental section. Furthermore, empirically, partitioned networks usually require more training iterations to converge.

**Performance**    Assuming the optimal hyper-parameters are given, training the full, non-partitioned networks based on those optimal values can be expected to yield better performance compared to the final model found by partitioned training. Partitioning the network inherently constrains the network capacity, causing some loss of performance. Opportunities for alleviating this performance loss while still enjoying single-run hyperparameter optimization through partitioned training will be left to future work. These include for example adjusting training rounds or increasing network capacity in the first place.

**Partitioning**    While partitioned networks allows for automatic optimization of, intuitively, hard to tune hyperparameters, such as augmentation parameters, they come with the additional limitation of requiring to partition both the data and the model. This introduces an additional hyperparameter, namely, the partitioning strategy. While our default strategy of assigning more parameters and data to the first chunk works reasonably well on all of the experiments we consider, if one targets obtaining the best possible performance on a given task, the partitioning strategy might need additional tuning. We provide some empirical results about the sensitivity to partitioning in appendix F.1

# I    EXPERIMENTAL DETAILS

## I.1    PARTITIONED TRAINING

**Partitioned parameter update scheduling**    The gradient computation of Equation 3, as described in the main text, requires that the data-points for updating a given subnetwork $w_s^{(k)}$ come from the appropriate dataset chunks $(x, y) \in \mathcal{D}_{1:k}$ for a chunk $k$. Depending on the partitioning scheme (Appendix C), evaluating different subnetworks for different chunks can or cannot be done in a single mini-batch. More specifically, the **random weight-partitioning** we chose for our experiments requires a separate mini-batch per subnetwork (in order to keep the memory cost the same as for standard neural network training). An immediate question arising from a chunked dataset and several partitions is to define the order and frequency of updates across subnetworks. In our experiments we define (non-uniform) splits of the training dataset $\mathcal{D}$ across the $C$ chunks, which requires a tailored approach to sampling the data. More specifically, for a given (normalized) ratio of chunk-sizes $[u_1, \ldots, u_C]$, each iteration of partitioned training proceeds as follows:

1. Sample a partition index $k \sim \text{Cat}(u_1, \ldots, u_C)$

2. Sample a mini-batch $\tilde{\mathcal{D}}$ of examples uniformly from $\mathcal{D}_{1:k}$.

3. Evaluate $\log p(\tilde{\mathcal{D}} | w_s^{(k)}, \psi)$ using subnetwork $w_s^{(k)}$ and

4. compute the (stochastic) gradient wrt. partition parameters $w_k$ (Eq. 3).

5. Update partition parameters $w_k$ using an optimizer, such as SGD or Adam.

This sampling scheme results in a data-point $(x, y) \in \mathcal{D}_k$ from earlier chunks to be sampled more often. Concretely, the probability that an example in chunk $k$ will be sampled is $\propto \sum_{i \leq k} u_i$. This is done so that each partition $w_k$ is updated with equal probability on each of the examples in $\mathcal{D}_{1:k}$ As a result, we use *with replacement* sampling for the partitioned network training throughout the experimental section.

**Gradient optimization of partitioned parameters**    A consequence of per-partition updates with the random weight partitioning scheme (appendix C) is that, for a chosen partition $\boldsymbol{w}_k$ to update, all other partitions do not receive a gradient update. In other words, the gradient at each iteration is sparse. Consequently, many off-the-shelve momentum-based optimizers will not account correctly. Specifically, we implement modifications to the PyTorch Paszke et al. (2019) provided optimizers that allow us to track per-partition momenta, number of steps, etc. Note that this creates a disconnect between the number of iterations across all partitions and the number of iterations per-partition. Doing so, however aligns the computational cost of training the partitioned network parameters with the cost of training regular neural network parameters. Regardless, we do not alter the way learning-rate schedulers behave in our experiments and anneal learning-rates according to the total number of iterations. Similarly, we report the total number of iterations when comparing against baselines that update all network-parameters per iteration.

While a simple gradient-accumulation scheme across mini-batches would result in a single gradient across all partitions, this approach inherently clashes with non-uniform partitioning $[u_1, \ldots, u_C]$. Instead, we chose to sequentially apply gradients computed on a single partition, as described in the previous paragraphs. A further advantage of this approach is that learning progress made by updating partition $\boldsymbol{w}_k$ immediately influences (and can improve) the prediction of subnetworks $\boldsymbol{w}_s^{(k)}, \boldsymbol{w}_s^{(k+1)}, \ldots, \boldsymbol{w}_s^{(C)}$.

**Gradient optimization of hyperparameters**    Our partitioned network scheme makes it easy to compute stochastic gradients of the hyperparameter objective $\mathcal{L}_{\mathrm{ML}}$ in Eq. 4 using batch gradient descent optimization methods. After every update to a randomly sampled network partition (see previous paragraph), we update hyperparamters $\psi$ as follows:

- sample a dataset chunk index $k \sim \mathrm{Cat}(\frac{u_2}{Z}, \ldots, \frac{u_C}{Z})$. Ratios are re-normalized to exclude $\mathcal{D}_1$.
- sample a mini-batch $\tilde{\mathcal{D}}$ of examples uniformly from $\mathcal{D}_k$ (Note the choice of $\mathcal{D}_k$ instead of $\mathcal{D}_{1:k}$).
- Evaluate $\log p(\tilde{\mathcal{D}}|\boldsymbol{w}_s^{(k-1)}, \psi)$ using subnetwork $\boldsymbol{w}_s^{(k-1)}$ and
- compute the (stochastic) gradient wrt. hyperparameters $\psi$ (Eq. 4).
- Update partition parameters $\psi$ using an optimizer, such as SGD or Adam.

The above sampling procedure yields an unbiased estimate of gradients in eq. 4.

The fact that we optimize hyperparameters with gradients based on data from a single chunk at a time is again a consequence of the random weight-partitioning scheme for the partitioned networks. It is possible to compute gradients wrt. $\psi$ for mini-batches with examples from multiple chunks at a time. With the random weight partitioning scheme, this would result in an increased memory overhead. Lastly, we could also accumulate gradients from different chunks, similarly to Immer et al. (2022), and this would likely result in a lower-variance estimate per update .

It is also possible to reduce the computational overhead of evaluating two mini-batches per iteration (one for updates to $\boldsymbol{w}_k$, one for $\psi$) as we do in our experiments by interleaving hyperparameter updates at less frequent intervals. We leave an exploration of these design choices to future work. Throughout all experiments, except those in the federated settings (see section J), we use the same batch-size for the hyperparameter udpates as for the regular parameter updates.

**Weight-decay**    For partitioned networks, whenever using weight-decay, we scale the weight decay for earlier partitions with the reciprocal of the number of examples in chunks used to optimize them, following the diagonal Gaussian prior interpretation of weight-decay. This makes the training compatible with the variational interpretation in Appendix B.

### I.2    PARTITIONED AFFINE TRANSFORMATIONS

In Appendix C we described how we realize partitioned versions of fully-connected and convolutional layers. Design choices for other parameterized network layers used in our experiments are described below.

**Normalization layers**   It is common-place in most architectures to follow a normalization layer (such as BatchNorm (Ioffe and Szegedy, 2015), GroupNorm (Wu and He, 2018)) with an element-wise or channel-wise, affine transformation. Namely, such a transformation multiplies its input $\boldsymbol{h}$ by a scale vector$\boldsymbol{s}$ and adds a bias vector $\boldsymbol{b}$: $\boldsymbol{o} = \boldsymbol{h} * \boldsymbol{s} + \boldsymbol{b}$. For random weight-partitioned networks, we parameterize such affine transformations by defining separate vectors $\{\boldsymbol{s}_1, \ldots, \boldsymbol{s}_C\}$ and $\{\boldsymbol{b}_1, \ldots, \boldsymbol{b}_C\}$ for each partition; the actual scale and bias used in a given subnetwork $\boldsymbol{w}_s^{(k)}$ are $\boldsymbol{s}_s^{(k)} = \prod_{i \in \{1, \ldots, k\}} s_i$ and $\boldsymbol{b}_s^{(k)} = \sum_{i \in \{1, \ldots, k\}} b_i$ respectively. This ensures that the final affine transformation for each subnetwork $\boldsymbol{w}_s^{(k)}$ depends on the parameters in the previous partitions $[1, \ldots, k-1]$. Doing so increases the parameter count for the partitioned networks in architectures that use those normalization layers by a negligible amount.

**Scale and bias in FixUp networks**   The FixUp paper (Zhang et al., 2019) introduces extra scales and biases into the ResNet architecture that transform the entire output of the layers they follow. We turn these into "partitioned" parameters using the same scheme as that for scales and biases of affine transformations following normalization layers.

For partitioned networks, through-out the paper, we match the proportion of parameters assigned to each partition $k$ in each layer to the proportion of data examples in the corresponding chunk $\mathcal{D}_k$.

## I.3   ARCHITECTURE CHOICES

**Input selection experiments**   We use a fully-connected feed-forward neural network with 2 hidden layers of size $[256, 256]$, and with GeLU (Hendrycks and Gimpel, 2016) activation functions. We initialise the weights using the Kaiming uniform scheme (He et al., 2015). For partitioned networks, we use the random-weight partitioning scheme.

**Fixup Resnet**   For all experiments using FixUp ResNets we follow Immer et al. (2022); Zhang et al. (2019), and use a 3-stage ResNet with channel-sizes $(16, 32, 64)$ per stage, with identity skip-connections for the residual blocks as described in He et al. (2016). The residual stages are followed by average pooling and a final linear layer with biases. We use 2D average pooling in the residual branches of the downsampling blocks.We initialize all the parameters as described in Zhang et al. (2019).

**Wide ResNet**   For all experiments using a Wide-ResNet-N-D (Zagoruyko and Komodakis, 2016), with N being the depth and D the width multiplier, we use a 3 stage ResNet with channel-sizes $(16D, 32D, 64D)$. We use identity skip-connections for the residual blocks, as described in He et al. (2016), also sometimes known as ResNetV2.

**ResNet-50**   We use the "V2" version of Wide ResNet as described in (Zagoruyko and Komodakis, 2016) and replace BatchNormalization with GroupNormalization using 2 groups. We use the 'standard' with with $D = 1$ and three stages of 8 layers for a 50-layer deep ResNet.

We use ReLU activations for all ResNet experiments throughout.

**MNIST CNN**   For the MNIST experiments, we use the same architecture as Schwöbel et al. (2021) illustrated in the replicated Table 9.

Table 9: CNN architecture for MNIST experiments

| Layer | Specification |
|---|---|
| 2D convolution | channels=20, kernel size=$(5, 5)$, padding=2, activation=ReLU |
| Max pooling | pool size=$(2, 2)$, stride=2 |
| 2D convolution | channels=50, kernel size=(5,5), padding=2, activation=ReLU |
| Max pooling | pool size=$(2, 2)$, stride=2 |
| Fully connected | units=500, activation=ReLU |
| Fully connected | units=50, activation=ReLU |
| Fully connected | units=10, activation=Softmax |

### I.4 TRAINING DETAILS

**Learning affine augmentations** For the parametrization of the learnable affine augmentation strategies, we follow prior works for a fair comparison. More specifically, for our MNIST based setup we follow the parametrization proposed in Schwöbel et al. (2021) whereas for our CIFAR10 based setup we use the generator parametrization from Immer et al. (2022).

**Input selection experiments** For the model selection (non-differentiable) input selection experiments, we train all variants with Adam with a learning rate of 0.001 and a batch-size of 256 for 10000 iterations. For both Laplace and partitioned networks, we do early stopping based on the marginal likelihood objective ($\mathcal{L}_{\mathrm{ML}}$ for partitioned networks). We use weight-decay 0.0003 in both cases. For the post-hoc Laplace method, we use the diagonal Hessian approximation, following the recommendation in (Immer et al., 2021). For partitioned networks, we divide the data and parameters into 8 chunks of uniform sizes. We plot results averaged across 3 runs.

**Mask learning for input selection experiment** We use the same optimizer settings as for the input selection experiment. We train for 30000 iterations, and optimize hyperparameters with Adam with a learning rate of 0.001. We divide the data and parameters into 4 uniform chunks.

**MNIST experiments** We follow Schwöbel et al. (2021), and optimize all methods with Adam with a learning rate of 0.001, no weight decay, and a batch-size of 200. For the partitioned networks and Augerino results, we use 20 augmentation samples. We use an Adam optimizer for the hyperparameters with a learning rate of 0.001 (and default beta parameters).

For Augerino on MNIST, we use the "safe" variant, as otherwise the hyperparameters and the loss diverge on every training run. We elaborate on this phenomenon in Appendix E. Otherwise, we follow the recommended settings from (Benton et al., 2020) and Immer et al. (2022), namely, a regularization strength of 0.01, and a learning rate for the hyperparameters of 0.05.

For both MNIST and CIFAR experiments, we found it beneficial to allocate more data to either the earlier, or the later, chunks. Hence, we use 3 chunks with $[80\%, 10\%, 10\%]$ split of examples for all MNIST and CIFAR experiments.

**CIFAR variations experiments** We again follow Immer et al. (2022), and optimize all ResNet models with SGD with a learning rate of 0.1 decayed by a factor of $100\times$ using Cosine Annealing, and momentum of 0.9 (as is standard for ResNet models). We use a batch-size of 250. We again use Adam for hyperparameter optimization with a learning rate of 0.001 (and default beta parameters). We train our method for $[2400, 8000, 12000, 20000, 40000]$ iterations on subsets $[1000, 5000, 10000, 20000, 50000]$ respectively for CIFAR-10, just as in (Immer et al., 2022). For all methods, we used a weight-decay of $1e-4$. For partitioned networks, we increase the weight decay for earlier partitions with the square root of the number of examples in chunks used to optimize them, following the diagonal Gaussian prior interpretation of weight-decay. We use 3 chunks with $[80\%, 10\%, 10\%]$ split of examples.

For RotCIFAR-10 results, we noticed our method hasn't fully converged (based on training loss) in this number of iterations, and so we doubled the number of training iterations for the RotMNIST results. This slower convergence can be explained by the fact that, with our method, we only update a fraction of the network parameters at every iteration.

**TinyImagenet experiments** Our experiments with TinyImagenet (Le and Yang, 2015) closely follow the setting for the CIFAR-10 experiments described above. Images are of size $64x64$ pixels, to be classified into one of 200 classes. The training-set consists of 100000 images and we compare our method against baselines on subset of $[10000, 50000, 100000]$ datapoints. For the standard version of TinyImagenet, we train for $[80000, 80000, 40000]$ steps respectively and for the rotated version of TinyImagenet we train for 120000 steps for all subset sizes. We tuned no other hyper-parameters compared to the CIFAR-10 setup and report our method's result for a partitioning with $[80\%, 20\%]$ across 2 chunks after finding it to perform slightly better than a $[80\%, 10\%, 10\%]$ split across 3 chunks in a preliminary comparison.

**Fine-tuning experiments**   For the fine-tuning experiments in table 2, we trained a FixUp ResNet-14 on a subset of 20000 CIFAR10 examples, while optimizing affine augmentations (following affine augmentations parameterization in (Benton et al., 2020)). We used the same optimizer settings as for all other CIFAR experiments, and trained for 80000 iterations, decaying the learning rate with Cosine Annealing for the first 60000 iterations. For fine-tuning of validation-set optimization models, we used SGD with same settings, overriding only the learning rate to 0.01. We tried a learning rate of 0.01 and 0.001, and selected the one that was most favourable for the baseline based on the test accuracy.

We also tried training on the full CIFAR-10 dataset, but found that all methods ended up within a standard error of each other when more than $70\%$ of the data was assigned to the first chunk (or training set, in the case of validation set optimization). This indicates that CIFAR-10 is sufficiently larger that, when combined with affine augmentation learning and the relatively small ResNet-14 architecture used, using the extra data in the 2nd partition (or the validation set) results in negligible gains.

## I.5   DATASETS

**Input selection synthetic dataset**   For the input selection dataset, we sample 3000 datapoints for the training set as described in section 5, and we use a fresh sample of 1000 datapoints for the test set.

**RotMNIST**   Sometimes in the literature, RotMNIST referes to a specific subset of 12000 MNIST examples, whereas in other works, the full dataset with 60000 examples is used. In this work, following (Benton et al., 2020; Immer et al., 2022) we use the latter.

## J   FEDERATED PARTITIONED TRAINING

In this section, we explain how partitioned networks can be applied to the federated setting, as well as the experimental details.

## J.1   PARTITIONED NETWORKS IN FL

In order to apply partitioned networks to the federated setting, we randomly choose a partition for each client such that the marginal distribution of partitions follows a pre-determined ratio. A given chunk $\mathcal{D}_k$ therefore corresponds to the union of several clients' datasets. Analogous to how "partitioned training" is discussed in the main text and Appendix I, we desire each partition $\boldsymbol{w}_k$ to be updated on chunks $\mathcal{D}_{1:k}$. Equation 3 in the main text explains which data chunks are used to compute gradients wrt. parameter partition $\boldsymbol{w}_k$. An analogous perspective to this objective is visualized by the exemplary algorithm in Figure 1 and asks which partitions are influenced (*i,e.*, updated) by data from chunk $\mathcal{D}_k$: A data chunk $\mathcal{D}_k$ is used to compute gradients wrt. partitions $\boldsymbol{w}_{k:C}$ through subnetworks $\boldsymbol{w}_s^{(k)}$ to $\boldsymbol{w}_s^{(C)}$ respectively. Consequently, a client whose dataset is assigned to chunk $\mathcal{D}_k$ can compute gradients for all partitions $\boldsymbol{w}_{k:C}$.

**Updating network partitions**   Due to the weight-partitioned construction of the partitioned neural networks, it is not possible to compute gradients with respect to all partitions in a single batched forward-pass through the network. Additionally, a change to the partition parameters $\boldsymbol{w}_k$ directly influences subnetworks $\boldsymbol{w}_s^{(k+1)}$ to $\boldsymbol{w}_s^{(C)}$. In order to avoid the choice of ordering indices $k$ to $C$ for the client's local update computation, we update each partition independently while keeping all other partitions initialised to the server-provided values that the client received in that round $t$: Denote $D_{i,k}$ as the dataset of client $i$ where we keep index $k$ to emphasize the client's assignment to chunk $k$. Further denote $\boldsymbol{w}_{j,i}^{t+1}$ as the partition $\boldsymbol{w}_j^t$ after having been updated by client $i$ on dataset $D_{i,k}$.

$$\boldsymbol{w}_{j,i}^{t+1} = \arg\max_{\boldsymbol{w}_j} \log p\left(D_{i,k} | (\boldsymbol{w}_1^t, \ldots, \boldsymbol{w}_j^t, \hat{\boldsymbol{w}}_{j+1}^t, \ldots, \hat{\boldsymbol{w}}_{j+C}^t), \psi\right) \quad \forall j \in [k, C], \qquad (25)$$

where the details of optimization are explained in the following section. We leave an exploration for different sequential updating schemes to future work. The final update communicated by a client to the server consists of the concatenation of all updated parameter partitions

$w_{.,i}^{t+1} = \text{concat}(w_{k,i}^{t+1}, \ldots, w_{C,i}^{t+1})$. Note that partitions $(w_1^t, \ldots, w_{k-1}^t)$ have not been modified and need not be communicated to the server. The resulting communication reductions make partitioned networks especially attractive to FL as data upload from client to server poses a significant bottleneck. In practice, we expect the benefits of these communication reductions to outweigh the additional computation burden of sequentially computing gradients wrt., to multiple partitions.

The server receives $w_{.,i}^{t+1}$ from all clients that participates in round $t$, computes the delta's with the global model and proceeds to average them to compute the server-side gradient in the typical federated learning fashion (Reddi et al., 2020).

**Updating hyperparameters**  The computation of gradients on a client $i$ wrt. $\psi$ is a straight-forward extension of equation 4 and the exemplary algorithm of Figure 1:

$$\nabla_\psi \mathcal{L}_{\text{ML}}(D_{i,k}, \psi) \approx \nabla_\psi \log p\left(D_{i,k} | w_{s,i}^{(t+1),(k-1)}, \psi\right), \qquad (26)$$

where $D_{i,k}$ corresponds to client $i$'s local dataset which is assigned to chunk $k$ and $w_{s}^{(t+1),(k-1)}$ corresponds to the $(k-1)$'th subnetwork after incorporating all updated partitions $w_{s,i}^{(t+1),(k-1)} = \text{concat}(w_1^t, \ldots, w_{k-1}^t, w_{k,i}^{t+1}, \ldots, w_{C,i}^{t+1})$. Note that we compute a full-batch update to $\psi$ in MNIST experiments and use a batch-size equal to the batch-size for the partitioned parameter updates for CIFAR10.

Upon receiving these gradients from all clients in this round, the server averages them to form a server-side gradient. Conceptually, this approach to updating $\psi$ corresponds to federated SGD.

## J.2 FEDERATED SETUP

**Non-i.i.d. partitioning**  For our federated experiments, we split the $50k$ MNIST and $45k$ CIFAR10 training data-points across 100 clients in a non-i.i.d. way to create the typical challenge to federated learning experiments. In order to simulate *label-skew*, we follow the recipe proposed in Reddi et al. (2020) with $\alpha = 1.0$ for CIFAR10 and $\alpha = 0.1$ for MNIST. Note that with $\alpha = 0.1$, most clients have data corresponding to only a single digit. For our experiments on rotated versions of CIFAR10 and MNIST, we sample a degree of rotation per data-point and keep it fixed during training. In order to create a non-i.i.d partitioning across the clients, we bin data-points according to their degree of rotation into 10 bins and sample using the same technique as for label-skew with $\alpha = 0.1$ for both datasets. Learning curves are computed using the $10k$ MNIST and $5k$ CIFAR10 validation data-points respectively. For the rotated dataset experiments, we rotate the validation set in the same manner as the training set.

**Architectures and experimental setup**  We use the convolutional network provided at Schwöbel et al. (2021) for MNIST and the ResNet-9 (Dys) model for CIFAR10 but with group normalization (Wu and He, 2018) instead of batch normalization. We include (learnable) dropout using the continuous relaxation proposed at Maddison et al. (2016) between layers for both architectures. We select 3 chunks for MNIST with a $[0.7, 0.2, 0.1]$ ratio for both, client-assignments and parameter-partition sizes. For CIFAR10, we found a $[0.9, 0.1]$ split across 2 sub-networks to be beneficial. In addition to dropout logits, $\psi$ encompasses parameters for affine transformations, *i.e.*, shear, translation, scale and rotation. We report results after $2k$ and $5k$ rounds, respectively, and the expected communication costs as a percentage of the non-partitioned baseline.

**Shared setting**  In order to elaborate on the details to reproduce our results, we first focus on the settings that apply across all federated experiments. We randomly sample the corresponding subset of $1.25k$, $5k$ data-points from the full training set and keep that selection fixed across experiments (*i,e.*, baselines and partitioned networks) as well as seeds. The subsequent partitioning across clients as detailed in the previous paragraph is equally kept fixed across experiments and seeds. Each client computes updates for one epoch of its local dataset, which, for the low data regimes of $1.25k$ data-points globally, results in single update per client using the entire local dataset. We averaged over 10 augmentation samples for the forward pass in both training and inference.

**MNIST & RotMNIST**  For $5k$ data-points and correspondingly 50 data-points on average per client, most clients perform a single update step. A small selection of clients with more than 64 data-points

performs two updates per round. For the experiments using the full dataset and a mini-batch size of $64$, each client performs multiple updates per round. After initial exploration on the baseline FedAvg task, we select a local learning-rate of $5e-2$ and apply standard SGD. The server performs Adam Reddi et al. (2020) with a learning rate of $1e-3$ for the model parameters. We keep the other parameters of Adam at their standard PyTorch values. We find this setting to generalize to the partitioned network experiments but found a higher learning rate of $3e-3$ for the hyper-parameters to be helpful. We chose the convolutional network from Schwöbel et al. (2021) with (learned) dropout added between layers. The model's dropout layers are initialized to drop $10\%$ of hidden activations. For the baseline model we keep the dropout-rate fixed and found $10\%$ to be more stable than $30\%$.

**CIFAR10 & RotCIFAR10**  We fix a mini-batch size of 32, leading to multiple updates per client per round in both, the full dataset regime as well as the $5k$ data-points setting. Similarly to the MNIST setting, we performed an initial exploration of hyperparameters on the baseline FedAvg task and use the same ones on partitioned networks. We used dropout on the middle layer of each block which was initialized to 0.1 for both the baseline and partitioned networks and whereas partitioned networks optimized it with $\mathcal{L}_{ML}$ and the concrete relaxation from Maddison et al. (2016), the baseline kept it fixed. For the server side optimizer we used Adam with the default betas and a learning rate of $1e-2$, whereas for the hyperparameters we used Adam with the default betas and a learning rate of $1e-3$. In both cases we used an $\epsilon = 1e-7$. For the local optimizer we used SGD with a learning rate of $10^{-0.5}$ and no momentum.

### J.3  MNIST LEARNING CURVES

In Figure 10 we show learning curves for the three considered dataset sizes on the standard MNIST task. Each learning curve is created by computing a moving average across 10 evaluations, each of which is performed every 10 communication rounds, for each seed. We then compute the average and standard-error across sees and plot those values on the y-axis. On the x-axis we denote the total communication costs (up- and download) to showcase the partitioned networks reduction in communication overhead. We see that especially for the low dataset regime, training has not converged yet and we expect performance to improve for an increased number of iterations.

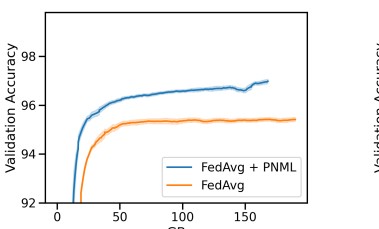 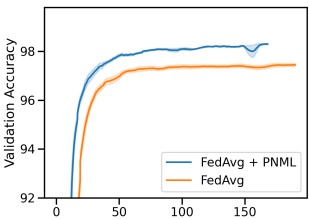 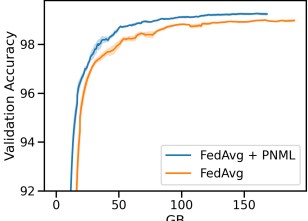

Figure 10: Learning curves for MNIST experiments on $1.25k$, $5k$ and $50k$ data-points respectively.

