# OpenReview forum: "Hyperparameter Optimization through Neural Network Partitioning"
_ICLR.cc/2023/Conference — ICLR 2023 poster_

### Official Review · Reviewer_Pdfa · 2022-10-20

**Confidence:** 3
**Correctness:** 3
**Technical Novelty And Significance:** 3
**Empirical Novelty And Significance:** 3
**Recommendation:** 6

**Clarity, Quality, Novelty And Reproducibility:**

The authors are very clear about how their method works and motivate its use very well. There is many prior work on the same topic but the authors discuss it in detail and clearly point out the difference. The idea itself reminds of the out-of-bag error used in bagging. However, its application to a single neural network and the theoretical discussion are novel.
I understand that the authors mostly use benchmarks considered in the most related work. I appreciate the interesting case study on federated learning. However, it would have been great if the authors went deeper into one particular problem and had a more extensive experiment. Many experiments seem to be toy examples. I would have loved to see a challenging hyperparameter optimization problem (jointly optimize augmentation, learning rate, etc.) including a comparison against state-of-the-art methods such as ASHA. Currently, it is unclear how good the hyperparameters actually are.

What is not fully clear to me is the difference to differentiable Laplace. Is the proposed method faster? If yes, how much faster. There is a claim that the proposed method is faster but I couldn't find any reported run times.

**Strength And Weaknesses:**

**Strengths**
- Good discussion of related work
- Diverse set of experiments
- Relevant problem, great motivation, well-written paper

**Weaknesses**
- Missing results to support the claim that the proposed method is computationally cheaper

**Summary Of The Paper:**

The authors present a method to tune hyperparameters which requires only a single training run and no validation data. This is achieved by partitioning the weights of a neural network and train each partition with respect to only its own training partition. Then, the remainder of the training data can be used for validation. The authors demonstrate the effectiveness of this approach for hyperparameter optimization in several scenarios including federated learning.

**Summary Of The Review:**

This is a well-written paper on an important scientific problem with important practical applications. The authors discuss related work in detail and demonstrate the effectiveness of the method on a couple of problems. Overall, I think the paper could be a great addition for the conference.

**After Rebuttal Phase**
Overall the paper was interesting and seemed novel to me, and therefore I was positive about it. However, based on the comments of some other reviewers, it might not have been very novel (I am not familiar with that related work) and therefore I've updated my confidence and recommendation accordingly.

---

> ### Author Response · Authors · 2022-11-14
> **Response to reviewer Pdfa**
>
> Dear reviewer, we would like to thank you for reviewing our work and providing encouraging comments. With the following, we aim to address them.
>
> ## Comparison against competitive hyperparameter optimization strategies
> For a comparison against a relatively strong baseline that is frequently used in practice, you can see our comparison against validation set optimization with finetuning. As we mentioned to reviewer Na31 as well, one of our primary goals was a computationally cheap method, which is particularly important in, e.g., federated learning. Therefore we opted to compare against similarly expensive methods and did not include grid-search-like methods such as ASHA.
>
> Furthermore, partitioned networks' simplicity allows them to be used in the inner loop of any off-the-shelf hyperparameter optimizer such as ASHA; for example, partitioned networks can be used to optimize augmentation parameters, whereas the ASHA outer loop can be used to optimize, e.g., the server learning rate in the federated setting without a validation set using our approximation to the marginal likelihood as an objective.
>
> ## Generalizability of hyperparameters
> > [...] Currently, it is unclear how good the hyperparameters actually are.
>
> At the moment, we are running experiments where we use partitioned networks to optimize the hyperparameters and, once those are optimized, we train from scratch using a non-partitioned network with those hyperparameters fixed. This would serve as a datapoint on how good the hyperparameters that partitioned networks learn are. We will report back once those experiments are completed.
>
> ## Computational efficiency of partitioned networks
> We agree that such a discussion is important to our story. We have updated our appendix to include both an asymptotic complexity analysis as well as empirical timings. We provide more details in the response to all reviewers.

---

> > ### Comment · Reviewer_Pdfa · 2022-11-24
> > **Thank you**
> >
> > Thanks a lot for your elaborate response. Is there any reason you fine tune on the validation set rather training from scratch again? Former will not work as well.

---

> > > ### Author Response · Authors · 2022-11-28
> > > **Re: Why fine-tune?**
> > >
> > > We wanted to showcase performance of methods that have similar computational cost. All the methods in Table 2 and 3 can reasonably be seen as optimising hyperparameters in a single training run; fine-tuning adds a moderate over-head compared to retraining from scratch.
> > > Fine-tuning will indeed likely perform worse than retraining.
> > >
> > > We could have included a comparison of partitioned networks against validation set optimization with retraining from scratch on the final set of hyperparameters found by both methods. This would allow us to compare the final hyperparameters found. However, we hope that the comparison against extensive random search in Appendix F with retraining gives comparable insight into how good the hyperparameters found by partitioned networks are.

---

> ### Author Response · Authors · 2022-11-24
> **2nd Response to reviewer Pdfa**
>
> ## How good are the hyperparameters found?
>
> We agree that whether the final hyperparameters found are good is an important part of the story.
> We performed additional experiments to explore this question during the rebuttal period.
> These confirm that 1) using the hyperparameters found with partitioned networks to retrain a "standard" neural network yields similar or better performance compared to that of a partitioned network, and 2) the performance of the retrained network is among the best achieved through a fairly extensive random search of the hyperparameter space.
> This indicates that the hyperparameters found by partitioned networks are indeed a good set of hyperparameters that translate well to standard neural networks, at least in the context of data augmentation learning.
> We added a new section "How good are the hyperparameters found?" (Appendix F) to the appendix to showcase these experiments.
>
> In more detail, we conducted an experiment where we performed a large random search over the hyperparameter space for the affine data augmentation learning task on the RotMNIST dataset. The results are shown in Figure 9 in the revised appendix. It showed that partitioned networks achieve performance that matches that of the standard neural network models with the best hyperparameters found through random search. Retraining a standard neural network on the hyperparameters found by partitioned networks retains this performance. For details, please see Appendix F.

---

> ### Author Response · Authors · 2022-12-12
> **Post-update Response to Reviewer [Pdfa]**
>
> We noticed that you decided to lower your rating based on some other reviewer's comments. We wanted to follow-up to attempt to clarify.
>
> From the explanation for the updated score, we understand that **if** the concerns about the novelty of this work were convincingly addressed, you believe that the paper is of a quality that deserves a score of **8**.
>
> Firstly, as far as we can tell, reviewer [Wybc] is the only one to question the novelty of our work. We **thoroughly disagree** with their verdict. In short, here is a list of things that we believe are novel about our method:
>
> - We believe that the idea of partitioning a single neural network, and associating the resulting subnetworks to chunks of data, is a novel and non-trivial innovation. It is with this innovation that one can efficiently optimize out-of-sample losses with a single network. Such losses can be losses stemming from approximations to the marginal likelihood, which is the case in our work, or even losses obtained from K-fold cross-validation. We provide a bit more details on the latter, in another comment. This partitioning is arguably at the core of our work. We are not aware of any work that does something similar in the literature.
>     - Furthermore, the initial conclusions about the novelty of our method by reviewer [Wybc] were based on an inaccurate understanding of our method as "a combination of K-fold cross-validation and dropout". We have since addressed that claim and reviewer [Wybc] agreed that our neural network partitioning is novel.
>
> - Explicitly training separate (sub-)networks on different subsets of data to estimate marginal likelihood is relatively novel, and has only been explored (to a limited extent, and in a different context) in [2]. We adopt the learning speed perspective on marginal likelihood from [2], but with our neural network partitioning, we can meaningfully use this objective throughout training. This allows us to successfully optimize neural network hyperparameters in a differentiable manner. This was *made* possible in our work through our novel partitioning scheme.
>   - We demonstrate this idea, when combined with our partitioned networks, can actually be used to optimize hyperparameters in a practical context. The work [2] was only demonstrated on toy tasks, and in the context of model selection.
>   - We show that our method works for model selection, learning invariances, hyperparameter optimization in Federated Learning and even outperforms validation-set optimization and an extensive random search, while being both computationally and memory efficient. To the best of our knowledge, no other marginal-likelihood-based method has been demonstrated to work for differentiably optimizing hyperparameters on the models and tasks we considered, without sacrificing scalability. As such, it should be of interest to the community, even solely on that ground.
>
>
> Based on these two points, we believe that we have thoroughly addressed any novelty concerns about our work. In case you think we didn't, or we missed something, please let us know and we could continue to discuss further.
>
> [2] Lyle, Clare, et al. "A bayesian perspective on training speed and model selection." Advances in Neural Information Processing Systems 33 (2020): 10396-10408.

---

> > ### Author Response · Authors · 2022-12-12
> > **Hypothetical Application of Partitioned Networks to K-fold Cross-validation**
> >
> > While this is something we did not explore in our work, one could form subnetworks $s_1 = (w_1,\hat{w}_2,\hat{w}_3),s_2 =(\hat{w}_1,w_2,\hat{w}_3),s_3=(\hat{w}_1, \hat{w}_2, w_3)$ to train the parameters on chunks $D_1, D_2, D_3$ respectively, where $\hat{w}_i$ are un-trainable parameters (set at initialization).
> >
> > Under such a training scheme, it is easy to form a K-fold CV loss by $CV_{loss} = \log p(D_1| \hat{w}_1, w_2, w_3) + \log p(D_2| w_1, \hat{w}_2, w_3) + \log p(D_3| w_1, w_2, \hat{w}_3)$. Neural network partitioning is the core that allows for efficient calculation of this CV-type of loss with a single neural network. In our work we focused on the out-of-sample loss as an approximation to marginal likelihood, but more K-fold-like type of losses is an interesting direction for future work.

---

### Official Review · Reviewer_ic1t · 2022-10-23

**Confidence:** 3
**Correctness:** 3
**Technical Novelty And Significance:** 3
**Empirical Novelty And Significance:** 2
**Recommendation:** 6

**Clarity, Quality, Novelty And Reproducibility:**

The paper is well written, and it contains many details on the methods, experiments setup, hyperparameters and training details, hence also good reproducibility. The quality is overall good, but I think the experiments could be more targeted towards the claim, as mentioned in the weakness section. The proposed method is novel and has a potential of big impact.

**Strength And Weaknesses:**

On the strength of the work, using marginal likelihood as a HPO objective is very interesting and I can see practical settings that marginal likelihood can be useful, such as in small data regime, federated learning or potentially a better objective for general HPO. On the methodology, it is a bold exploration that partitioning the networks into subnetworks and training them separately over data chunks. On the empirical evaluations, it covers a wide range of tasks from different areas. It is also a well written paper, with clear structures and abundant details for reproducibility.

On the weakness, my first concern is motivation. The author mentioned partitioning the parameters, in the beginning of page 4, “can be a reasonable approximation in function space”. Why is that? Also, one thing I am not sure, if the goal is to approximate eq(2) and avoid the model sees every datapoint in the training set after 1 epoch, why one can’t create $C$ copies of the whole parameters and assign each copy to one $D_k$, without even partitioning the parameter (update all the parameter with related data chunk in each copy)?

Then it is the support for the claims. The authors argued the proposed method “is more scalable than previous works that rely on marginal likelihood and Laplace approximations”. I can see why it is more scalable (avoid inverting a Hessian (Immer et al., 2021)), but I don’t see any empirical results supporting it.

Since the main contribution is a new method to approximate the marginal likelihood, the comparison to relevant baselines, for example, Schwobel et al. (2021) and Immer et al., (2022), will be critical to demonstrate its usefulness. But they are not all the times included in all the experiments. For example, in the experiments of learning invariances through data-augmentations Schwobel et al. (2021) is not in Table 1 and Immer et al., (2022) not in Fig 3(a)? There are no baselines (Schwobel et al. (2021) and  Immer et al., (2022)) in the experiments for “Comparisons to traditional training / validation split”. It is hard to attribute the improvement to the partitioned network (a particular way to approximate learning speed) or the learning speed objective in general. Similarly, no other HPO baselines are included in the Federated Learning experiments. Are there some particular reasons why these results are missing? If so, it would be nice to explain them in the paper.

In the end, I am wondering about the impact of partitioning? How sensitive are the results to a partition?


=== After reading authors' response ===

The authors' response answered all my questions and I increased the score to 6.

**Summary Of The Paper:**

The work starts with using marginal likelihood as an alternative objective to validation metric for HPO, which has advantages in the small data region and potentially better generalization performance. It then talked about how to compute marginal likelihood and proposed an efficient method that only requires a single training run and no validation set. The experiments on synthetic and a diverse set of real world experiments (data argumentation, classical HPO setting and Federated Learning) demonstrated its usefulness.

**Summary Of The Review:**

The paper has many merits and they are mentioned already in the strength section. But the  weaknesses, so far, are also clear from my perspective. Thus, without more clarifications or justifications for the proposed method, I am not confident to accept the paper.

---

> ### Author Response · Authors · 2022-11-14
> **Response to Reviewer ic1t [1 / 3]**
>
> Dear reviewer, we would like to thank you for your thorough review. We hope that the following will address your remaining concerns.
>
> ## Sparse networks can yield reasonable function space approximation
> > The author mentioned partitioning the parameters, in the beginning of page 4, “can be a reasonable approximation in function space”. Why is that?
>
> As other reviewers also had questions about this intuition, we have provided a response in the general global comment. We have also further clarified this point in the paper.
>
> ## Learning speed objective / partitioned networks distinction
> > [...] if the goal is to approximate eq(2) and avoid the model sees every datapoint in the training set after 1 epoch, why one can’t create C copies of the whole parameters and assign each copy to one Dk, without even partitioning the parameter (update all the parameter with related data chunk in each copy)?
>
> Yes, that is absolutely possible! The practical limitation with this approach is the need to store and simultaneously train $C$ copies of the model (since the hyperparameters need to be synchronised at each iteration). For larger models, which might barely fit on the GPU, this is a significant limitation. With partitioned networks, we wanted to explore whether we can sidestep this limitation.
>
> For models for which we can fit multiple copies of on a single GPU, we added the results for the learning speed objective computed in this manner as a baseline. As expected, such a setting corresponds to an upper bound for partitioned networks, although the gap appears to be quite small.
>
> ## Scalability
> As other reviewers also asked for more details about our claim for scalability, we have provided a response in the general global comments, and added details to the paper (Appendix C in the revised version in particular).

---

> > ### Author Response · Authors · 2022-11-14
> > **Response to Reviewer ic1t [2 / 3]**
> >
> >
> > ## Comparisons to prior work
> > We also want to go into more detail on why each method was or wasn't showcased as a baseline in each experiment.
> >
> > We only include the method by Schwobel et al. (2021) [1] in the MNIST baselines, since — as the authors of that work elaborate themselves — their method doesn't scale to CIFAR10. Section 7 of their paper explores in detail the failed attempt of adapting Last Layer Marginal Likelihood to work on CIFAR10. We added an elaboration on this in the new Appendix section on scalability (Appendix C in the revised version).
> >
> > As for lack of Differentiable Laplace in the subsection exploring the comparisons between validation set optimization and partitioned networks, the purpose of this comparison was a bit different. We wanted to compare to validation set optimization, since that's one other methodology that also partitions/chunks the data in some sense. Since there is no notion of chunk size in Differentiable Laplace, we don't think that comparison would be fitting in Table 2.
> >
> > The reviewer also pointed out that we didn't include any of the other baselines in the federated experiments.
> > Since Augerino can be trivially generalized to the federated setting, we added it as a federated baseline in the paper in Table 4. Overall, we observe that FedAvg with Augerino improves upon vanilla FedAvg in some cases, however, it is lagging behind FedAvg with our partitioned networks method. We go into more detail on why we didn't federate the other methods below.
> >
> > **Federating Differentiable Laplace.** Although it might be possible to adapt the Differentiable Laplace [2] baseline to the Federated Learning setting, the extension is non-trivial. Their method requires computing the (layerwise Kronecker-factored) Hessian of an objective on the entire dataset before updates to the hyperparameters can be made. This would require designing a federated schedule which alternates between regular training, computing and aggregating the Hessian, and updating the hyperparameters. This is a non-trivial extension, both conceptually and (especially) implementation-wise, and would arguably constitute a new method.
> >
> > As such, we would consider designing a Federated variant  of Differentiable Laplace to be beyond the scope of this work.
> >
> > We also note that such a federated method likely wouldn't make much sense in practice (without some major innovations at least), as aggregating a (even Kronecker factored) Hessian on the entire federation would be impractical in the typical target settings for Federated Learning methods with many clients and low-power mobile devices.
> >
> > **Federated Last Layer Marginal Likelihood.** Last layer marginal likelihood requires a non-trivial schedule for how the parameters of the neural network and the Gaussian Process (GP) fitted on top of it are optimized. Without it, the authors claim that the neural network parameters overfit, and thus the gradient signal for learning augmentations through the GP likelihood is not informative. Adapting such a schedule to the federated setting requires additional considerations that we would consider beyond the scope of this work. Finally, as mentioned above, their method doesn't scale to CIFAR10 even in the centralized setting.

---

> > > ### Author Response · Authors · 2022-11-14
> > > **Response to Reviewer ic1t [3 / 3]**
> > >
> > >
> > > ## Sensitivity to partitioning
> > > This is something we indeed acknowledge as a limitation. We have revised our appendix to include both a discussion on it as well as an ablation study on MNIST that uses varying numbers of partitions and assignments of parameters to partitions. Empirically, on MNIST, the differences in final performance appear to be small. However, the differences are still there, so if the goal is to maximise performance, the partitioning strategy should be adjusted.
> > >
> > > ---
> > > Please let us know if you have any other comments, feedback, or if there is anything else we can clarify!
> > >
> > > ---
> > > [1] Schwöbel, Pola, et al. "Last Layer Marginal Likelihood for Invariance Learning." International Conference on Artificial Intelligence and Statistics. PMLR, 2022. https://proceedings.mlr.press/v151/schwobel22a/schwobel22a.pdf
> > >
> > > [2] Immer, A., van der Ouderaa, T. F., Fortuin, V., Rätsch, G., & van der Wilk, M. (2022). Invariance Learning in Deep Neural Networks with Differentiable Laplace Approximations. arXiv preprint arXiv:2202.10638. https://arxiv.org/pdf/2202.10638.pdf

---

> > > > ### Comment · Reviewer_ic1t · 2022-11-23
> > > > **Reply to authors' response**
> > > >
> > > > I want to thank the authors for their detailed response. My questions are all answered but I would still suggest the author to discuss the limitation not only in appendix but also in the main text. I will increase my score to 6.

---

> ### Author Response · Authors · 2022-12-12
> **Post-rebuttal response - Thanks**
>
> We appreciate that you revised your score. We will make sure to address the remaining concern — namely, mentioning the limitations in the main text — in our next revision.

---

### Official Review · Reviewer_Na31 · 2022-10-25

**Confidence:** 3
**Clarity, Quality, Novelty And Reproducibility:** 1. Originality

Although I am not ver…
**Correctness:** 3
**Technical Novelty And Significance:** 3
**Empirical Novelty And Significance:** 2
**Recommendation:** 6

**Strength And Weaknesses:**

Strength
- The paper is well-written and clearly motivated. I believe that the idea of explicit network partitioning is interesting and novel and the work is relevant to the ICLR community.
- The technical details of the paper look correct.
- The experiment setup is detailed making it easy to implement and reproduce the results.

Weakness
- While the paper argues that the proposed method is more efficient compared to previous methods such as Laplace approximation, it is not empirically shown in Section 5. I am happy to increase the score if this concern is addressed.
- The proposed method is not compared with a competitive hyperparameter optimization baseline (that uses a validation set). While I understand that the proposed framework does not require partitioning the train and validation set, it would be helpful to understand how the method compares to other strong hyperparameter optimization baselines.
- The authors do not sufficiently describe the limitation of the proposed approach. What are some additional hyperparameters and how sensitive are they? Can the method be more scalable (in terms of both parameter and hyperparameter) compared to the baseline methods?

**Summary Of The Paper:**

The paper proposes an algorithm that partitions both the data and model parameters to efficiently approximate the log marginal likelihood. The key idea is to define the out-of-training-sample loss with the partitioned network and utilize this objective for hyperparameter optimization. Empirically, the authors show that the proposed method can adapt various hyperparameters in neural networks without a validation set. Moreover, the proposed methods can be utilized in federated learning, where cross-validation is difficult.


**Summary Of The Review:**

Overall, I believe that the idea presented in this paper is interesting and novel and the paper is well-written. While I have concerns about the diversity of the baseline in the experiment section and the lack of computation and timing experiments. At the current state, I recommend 6.

---

> ### Author Response · Authors · 2022-11-14
> **Response to Reviewer Na31**
>
> We would like to begin by thanking you for taking the time to review our work. We greatly appreciate that you find our work well-written, clearly motivated, interesting and novel. We hope that the following will address your remaining concerns and you will consider revising your score.
>
> ## Computational efficiency analysis
> As other reviewers also raised the concern of computational efficiency of our method relative to prior work, we have provided our response in a global comment. In brief, we have revised our appendix with both timings as well as a complexity analysis of partitioned networks and how they fare against Benton et al. and Immer et al.
>
> ## Comparison against competitive hyperparameter optimization baselines
> We believe that the baseline we compare against in Table 2 is fairly competitive and close to what is frequently done in practice. There, separate training and validation sets are created which are used in a traditional fashion; the training set is used to optimize the parameters of the (non-partitioned) model, whereas the validation set is used in order to measure the validation loss and compute the gradients with respect to the (augmentation) hyperparameters. To further strengthen our baseline, we also show the performance when further considering finetuning the network on the joint training and validation data as soon as the hyperparameters have been optimized. Partitioned networks have similar or better performance to such a baseline. Since one of our main motivations is computational efficiency, we chose this baseline as it has similar computational costs. Having said that, partitioned networks are orthogonal to more expensive methods that rely on, e.g., grid-search to optimize the hyperparameters; we can use the partitioned approximation to the marginal likelihood as the objective in any off-the-shelf hyperparameter optimization method.
>
> ## Limitations of our method
> We agree that we have not discussed the limitations of partitioned networks sufficiently. To that end, we have added one more section in our appendix where we discuss them in detail. We also now summarize them in the conclusion and point to the appendix. To briefly summarize it here, partitioned networks usually require a bit more training time and have a slight decrease in performance (when hyperparameters are not optimized) due to restricting the final network's capacity, i.e., a fully retrained network on the final hyperparameters usually has better performance. In addition, partitioned networks also introduce an extra hyperparameter, namely, the partitions. Having said that, the strategy we used in the main paper seems to be fairly robust (unless explicitly stated, we use the same strategy for all experiments in our submission).

---

> ### Author Response · Authors · 2022-12-12
> **Follow-up on timing experiments**
>
> We were in contact with the authors of the Differentiable Laplace paper [1], and with their help, we worked out the issues that allowed us to run their method on our MNIST architecture. We empirically measured that the marginal-likelihood hyperparameter update on the MNIST CNN takes $2015.6\times$ longer than a regular parameter update. This is likely due to the cubic scaling of their method with the width of the network, as the network used was relatively wide. We will update the respective entry in Table 8 in Appendix C in the empirical timings comparison.
>
> [1] Immer, Alexander, et al. "Invariance Learning in Deep Neural Networks with Differentiable Laplace Approximations." arXiv preprint arXiv:2202.10638 (2022).

---

> ### Author Response · Authors · 2022-12-12
> **Follow-up**
>
> We wanted to follow-up on the questions relating to demonstrating the scalability of our method. In particular, you noted:
>
> > While the paper argues that the proposed method is more efficient compared to previous methods such as Laplace approximation, it is not empirically shown in Section 5. I am happy to increase the score if this concern is addressed.
>
> We hope that, with the empirical and theoretical analysis we added during the rebuttal period, we thoroughly address any concerns with respect to efficiency of our method. With that, we would appreciate if you could consider revising your score. If not, let us know why so that we can further improve our work.

---

### Official Review · Reviewer_Wybc · 2022-10-29

**Confidence:** 3
**Correctness:** 3
**Technical Novelty And Significance:** 2
**Empirical Novelty And Significance:** 2
**Recommendation:** 5

**Clarity, Quality, Novelty And Reproducibility:**

Novelty: Mentioned in main review, but it is limited. It is extremely unclear if the main benefit of the method is data efficiency, or improved generalization, or better optimization runtime, or all tree.

Clarity/Quality: Generally good but there are some concerns:

It is mentioned that: Although a subset of the parameters in each w (k) s is fixed and hence would likely be a poor approximation to the posterior over the weights, it can be a reasonable approximation in function space. Why would this be true?

It is mentioned that: "There is no need to compute nor invert a Hessian with respect to the weights, as in the Laplace approximation". Why would this hold since the posterior for most Bayesian networks are still intractable to compute?

In Figure 1: Updated the caption to explain why hat(w_i) is fixed.

In Figure 2: Why do the different line colors mean? Why are there no other baselines compared here?


**Strength And Weaknesses:**

The paper mainly proposes a technique of splitting the training/validation dataset in an iterative manner, inspired by the marginal likelihood decomposition for cross validation, which is claimed to be an optimization objective that requires no validation data.

While marginal likelihood is standardly used for Bayesian model selection, the choice of Bayesian prior/function class strongly influences the generalization capabilities of these methods since almost all loss functions, such as the L2/cross entropy can be derived by using Bayesian priors. In fact, a recent paper [https://arxiv.org/abs/2202.11678] demonstrated the pitfalls of using marginal likelihood without validation. Since the marginal likelihood is the main theoretical justification for the method design, I find the justification extremely weak so giving a theoretical example with concretely define Bayesian priors/assumptions would be better.

The novelty of the paper is also limited in the practical algorithm. The method of partitioning the model and parameters are reminiscent of cross validation and drop out techniques, which have both been readily used in the ML community. Adding a baseline of using dropout + cross validation would be better, as well as spending more time of using the toy example (of 1/2 spurious variables) to illustrate WHY your method does better than all the "standard" baselines.

**Summary Of The Paper:**

The main contribution is to tune differentiable “hyperparameters” by splitting the trainable parameters and dataset in a way that separates the data seen for tuning hyperparameters, ensuring that the set of hyperparameters are generally good for a wide range of parameters. The splitting technique is inspired by the marginal likelihood.

**Summary Of The Review:**

The main method proposed in the paper is limited in its novelty and shaky in its theoretical groundings. The paper does not clearly state the main benefits of using this method and such empirical benefits are not convincingly displayed in the experiments, especially in the "toy model" which are designed to clearly demonstrate the superiority of the new method over baselines.

---

> ### Author Response · Authors · 2022-11-14
> **Response to Reviewer Wybc [1 / 2]**
>
> Dear reviewer, we would like to thank you for reviewing our work.
>
> ## Clarifications on our method
> At the beginning of training, we partition both the training data and parameters, such that we can form subnetworks that are trained on specific subsets of data. This partitioning is done once and kept fixed throughout training. This is done in order to have an easy and "any-time" way to measure the loss of each subnetwork on unseen (for that specific subnetwork) chunks of training data. The latter constitutes the objective we use to optimize the hyperparameters, which guides them to be optimized in way that allows for good out-of-sample behaviours of the subnetworks (instead of being good for a wide range of parameters).
>
> ## Marginal likelihood as an objective
> We agree that, as shown by Lofti et al. (2022) [1], marginal likelihood is sensitive to the choice of the prior distribution. In that work, they also put forward *conditional marginal likelihood* as an adjusted objective, and showcase it alleviates most of their concerns with standard marginal likelihood. This is precisely why in Eq. 5 (where we present the hyperparameter gradient), we omit the contribution of the first subnetwork to the hyperparameter gradient. By omitting this term, we are essentially computing the gradient of the hyperparameters $\psi$ with respect to (an approximation to) $\log p(D_{2:K}|D_1, \psi)$, i.e., the *conditional marginal likelihood*. Conditioning on $D_1$, allows for using a more informative prior (which in our case is the first subnetwork). As the paragraph underneath Eq. 5 explains, this objective better correlates with generalization performance, and is precisely the one suggested by Lotfi et al. [1] (which both you and us cite).
>
> ## Novelty of our approach
> We respectfully disagree that the novelty of our approach is limited. Partitioning the network and data in a way that allows for computational, memory efficient and any-time measurement of the generalization error for optimizing the hyperparameters has not been proposed before. We would like to point to other reviewers' assessments on that front:
>
> > Overall, I believe that the idea presented in this paper is interesting and novel and the paper is well-written.
>
> > The proposed method is novel and has a potential of big impact.
>
> > The idea itself reminds of the out-of-bag error used in bagging. However, its application to a single neural network and the theoretical discussion are novel.
>
> While it is reminiscent of cross-validation and dropout, it differs in critical ways. Firstly, K-fold cross-validation requires $K$ independent training runs and models,  which, especially in the deep learning and federated learning regimes, is impractical. In contrast, partitioned networks require just a single run and model.
>
> Secondly, dropout samples a fresh mask for the network on every iteration and datapoint in the batch, which, in contrast to partitioned networks, does not allow for any-time measurement of the generalization error for hyperparameter optimization. The purpose of dropout is different; it is a way to regularize the model. The dropout rates are frequently considered as hyperparameters and, in fact, are hyperparameters we can and do optimize with partitioned networks in our federated learning experiments. Since the purpose of dropout is different, we cannot immediately envision how a baseline of cross-validation and dropout would work. It would be helpful if the reviewer could further elaborate.
>
> Finally, since partitioned networks are computationally cheap and efficient, the purpose of the toy task was to show that they perform at least on par with more expensive methods, such as the Laplace approximation to the marginal likelihood. Besides that, we can also see that partitioned networks have comparable or better performance on all the tasks we consider, while still being computationally cheap.
>
> ## Intuitions about function space approximation
> Since other reviewers also had questions about this intuition, we have provided a response in the general global comment. We have also further clarified this point in the paper.

---

> > ### Author Response · Authors · 2022-11-14
> > **Response to Reviewer Wybc [2/2]**
> >
> >
> > ## Claims about the Hessian
> > The majority of works that employ (conditional) marginal likelihood optimization for various hyperparameters employ the Laplace approximation to the posterior distribution over the weights of the neural network, since the true posterior (as rightly pointed out by the reviewer) is intractable. The Laplace approximation to the posterior is a multi-variate Gaussian distribution, where the mean is point-estimated with standard maximum-a-posteriori training and the covariance is (usually an approximation to) the inverse Hessian of the loss function around that point. This expensive approximation to the posterior is _necessary_ to compute the hyperparameter objective in the prior works we compare to [2,3], as that objective directly depends on the Hessian. This is where the main computational challenges arise  (and it was one of the areas where the work of Immer et al. [3] improved upon prior work).
> >
> > There is no need to compute the Laplace approximation to the posterior for our proposed formulation of the hyperparameter objective — a point-estimate approximation is compatible with our objective. As such, the partitioned networks do not employ any Laplace approximations, and hence avoid that extra (significant) computational cost. You can find more information about the computational complexity comparisons of partitioned networks to other related works in our revised appendix.
> >
> > ### What's the prior?
> > As the reviewer correctly pointed out, standard neural network training with weight-decay can be interpreted as finding a maximum-a-posteriori point-estimate under an isotropic Gaussian prior. The reviewer enquired about what prior did we use; those details can be found in appendix on "Training Details" in paragraphs title "CIFAR Variations Experiments" and "MNIST Experiments". In short, we match the weight-decay to our baselines, which in the case of CIFAR experiments corresponds to a $\mathcal{N}(0, 0.45^2)$ Gaussian prior (on the full dataset). On MNIST, the baselines do not use weight-decay; in order to match, we effectively use an improper "uniform" prior.
> >
> > ## Minor clarity comments:
> > In Figure 2, the legend is integrated into the label of the appropriately coloured y-axis. To ensure clarity, we added an explanation of the colours in the caption.
> >
> > ---
> >
> > We hope that our response has addressed your concerns, and thus you could consider revising your score. Let us know if you have any other comments or feedback.
> >
> > ---
> >
> >
> > [1] Sanae Lotfi, Pavel Izmailov, Gregory Benton, Micah Goldblum, Andrew Gordon Wilson, "Bayesian Model Selection, the Marginal Likelihood, and Generalization", 2022, https://arxiv.org/pdf/2202.11678.pdf
> >
> > [2] Immer, A., Bauer, M., Fortuin, V., Rätsch, G., & Emtiyaz, K. M. (2021, July). Scalable marginal likelihood estimation for model selection in deep learning. In International Conference on Machine Learning (pp. 4563-4573). PMLR. https://arxiv.org/abs/2104.04975
> >
> > [3] Immer, A., van der Ouderaa, T. F., Fortuin, V., Rätsch, G., & van der Wilk, M. (2022). Invariance Learning in Deep Neural Networks with Differentiable Laplace Approximations. arXiv preprint arXiv:2202.10638. https://arxiv.org/pdf/2202.10638.pdf

---

> > ### Comment · Reviewer_Wybc · 2022-11-17
> > **Similar to K-fold validation**
> >
> > I think my main reservation is that your method is similar to a sequential k-fold validation approach. You mentioned that K-fold cross-validation requires  independent training runs and models, but you can use a K-fold validation sum loss, by first summing across all validation losses. However, I do think that splitting the training parameters is interesting and I've updated my score.

---

> > > ### Author Response · Authors · 2022-11-24
> > > **2nd Response to reviewer Wybc**
> > >
> > > We'll try to respond to each bit in turn. Firstly:
> > >
> > > >  [...] I do think that splitting the training parameters is interesting and I've updated my score.
> > >
> > > We appreciate that you read through our rebuttal and considered updating your score!
> > >
> > > > I think my main reservation is that your method is similar to a sequential k-fold validation approach.
> > >
> > > We absolutely agree that the objectives in equations (2) and (3) are very reminiscent of cross-validation. That's an intrinsic property of the marginal likelihood that we find interesting, and try to emphasize in the paper.
> > >
> > > There is something unique about this type of cross-validation that comes from decomposing the marginal likelihood — namely, where we fix the ordering over the shards, and a model evaluated on any given shard is trained on all preceding shards — that allows us to partition the network.
> > > This allows us to re-use the weights fit to data shards $D_{1:k-1}$ in the sub-network used for fitting $D_{1:k}$
> > >
> > > > You mentioned that K-fold cross-validation requires independent training runs and models, but you can use a K-fold validation sum loss, by first summing across all validation losses.
> > >
> > > If we understood this properly, the proposal is: after partitioning the data into a validation and train set in $K$ different ways $((D^1_{train}, D^1_{val}), \dots, (D^K_{train}, D^K_{val}))$, train a separate neural network model on each of the train sets $D^i_{train}$ and sum the validation losses on respective $D^i_{val}$?
> > > I think that's similar to the baseline reviewer \[ic1t\] inquired about.
> > > This would still require training $K$ models in parallel (since the hyperparameters need to be synced across models after every hyperparameter update) to compute the hyperparameter objective. We have included this way of computing the objective in equation (2) as a baseline for models where we can fit $K$ copies of the neural network in memory.

---

> ### Author Response · Authors · 2022-12-12
> **Summary of responses to to Reviewer Wbyc on novelty claims**
>
> We wanted to again follow-up on the discussion, and summarize our claims and contributions.
>
> Firstly, we still *thoroughly disagree* with the limited novelty verdict. In short, here is a list of things that we believe are novel about our method (assimilated from response to reviewer [Pdfa]):
>
> - We believe that the idea of partitioning a single neural network, and associating the resulting subnetworks to chunks of data, is a novel and non-trivial innovation. It is with this innovation that one can efficiently optimize out-of-sample losses with a single network. Such losses can be losses stemming from approximations to the marginal likelihood, which is the case in our work, or even losses obtained from K-fold cross-validation. We provide a bit more details on the latter, in another comment to reviewer [Pdfa]. This partitioning is arguably at the core of our work. We are not aware of any work that does something similar in the literature. You agreed that our neural network partitioning is interesting.
> - Explicitly training separate (sub-)networks on different subsets of data to estimate marginal likelihood is relatively novel, and has only been explored (to a limited extent, and in a different context) in [2]. We adopt the learning speed perspective on marginal likelihood from [2], but with our neural network partitioning, we can meaningfully use this objective throughout training. This allows us to successfully optimize neural network hyperparameters in a differentiable manner. This was *made* possible in our work through our novel partitioning scheme.
>   - We demonstrate this idea, when combined with our partitioned networks, can actually be used to optimize hyperparameters in a practical context. The work [2] was only demonstrated on toy tasks, and in the context of model selection.
>   - We show that our method works for model selection, learning invariances, hyperparameter optimization in Federated Learning and even outperforms validation-set optimization and an extensive random search, while being both computationally and memory efficient. To the best of our knowledge, no other marginal-likelihood-based method has been demonstrated to work for differentiably optimizing hyperparameters on the models and tasks we considered, without sacrificing scalability. As such, it should be of interest to the community, even solely on that ground.
>
>
> Based on these two points, we believe that we have thoroughly addressed any novelty concerns about our work raised here.
>
> [2] Lyle, Clare, et al. "A bayesian perspective on training speed and model selection." Advances in Neural Information Processing Systems 33 (2020): 10396-10408.

---

### Author Response · Authors · 2022-11-14
**Response to all reviewers [1 / 3]**

Dear reviewers, we would like to thank you for your positive comments, constructive feedback and tips to improve our submission.

We especially want to highlight that the reviewers judged our work **relevant to the ICLR community** (*Na31*), that it **could be a great addition for the conference** (*Pdfa*) and an **important scientific problem with important practical applications** (*Pdfa*). Reviewer *ic1t* welcomed our methodolgy as a **bold exploration**.

They appreciated it as **interesting** (*Na31*, *ic1t*), its contribution as **novel** (*Na31*, *Pdfa*), and valued our provided level of detail (*Na31*,*ic1t*), judging it **abundant** (*ic1t*) in enabling **reproducability**.

Reviewers *ic1t* and *Pdfa* highlighted the disversity of our empirical evaluation, while reviewer *Pdfa* also especially appreciated our discussion of related work.
Finally, three reviewers (*ic1t*, *Na31*, *Pdfa*) mentioned that our work is well-written.

***

In this top-level comment we'd like to address commonly raised points. We will address reviewer-specific feedback in separate comments to individual reviews.

---

> ### Author Response · Authors · 2022-11-14
> **Response to all reviewers [2 / 3]**
>
>
> ## Computational complexity and memory requirements
> All reviewers asked for substantiation of our claims to improved computational complexity and reduced memory requirements in relationship to the Laplace approximation (Immer et al., 2022). We address these in a new appendix section in the revised version of the paper and give a summary here. Please check the revised paper for a more detailed discussion.
>
> >Please note that we chose not to re-implement their method as it is rather involved and their code-base was published only after the ICLR submission deadline.
>
> >Further note that at the time of this post, the official codebase for Differentiable Laplace <https://github.com/tychovdo/lila> (which was released after our initial submission) crashes with out-of-memory error on our 10GB GPU. We contacted the authors and will update this discussion and the paper should the authors respond in time.
>
> Following (Immer et al., 2022) we first provide a complexity analysis in big-$O$ notation.
> Our notation is as follows
>
>  Symbol | Interpretation
>   -------------------- | :-----------------------
>  $N$ | Number of datapoints in dataset $\mathcal{D}$
>  $N_B$ | Batch size
>  $S$ | Number of augmentation samples (for invariance learning, else $S=1$)
>  $C$ | Output size (number of classes)
>  $D$ | Feedforward network layer max. width
>  $L$ | Feedforward network depth
>  $P$ | Number of parameters (s.t. $O(P) = O(LD^2 + DC)$)
>
>
>
> ### Computational complexity
>
>  Method | Parameter Backward | Hyperparameter Backward | Hyperparameter Objective
>  -------------------- | :----------------------- | :------------- | :-------------
>  Partitioned Networks | $O(N_BPS)$ |  $O(N_BPS)$ |  $O(N_BPS)$
>  Augerino | $O(N_BPS)^*$ |  $O(N_BPS)^*$ |  $O(N_BPS)$
>  Differentiable Laplace | $O(N_BPS)$ |  $O(N_BPS + NCP + NCDLS + LD^3)^{**}$ |  $O(NPS + NCP + NCDLS + LD^3)$
>
> >$^*$ Augerino computes parameter and hyper-parameter gradients in a single backward pass
>
> >$^{**}$ Part of this computation can be amortized over multiple backward passes
>
> In terms of the big-${O}$ notation, our method has the same computation complexity for updating network- and hyperparameters compared to Augerino, while Differentiable Laplace's complexity scales cubically with network width. Note here that the maximum-width network the authors consider is of width $256$.
>
> ### Memory complexity
>
>  Method | Parameter Backward | Hyperparameter Backward | Hyperparameter Objective
>  -------------------- | :----------------------- | :------------- | :-------------
>  Partitioned Networks | $O(N_BSLD + P)$ |  $O(N_BSLD + P)$ |  $O(N_BSD + P)$
>  Augerino | $O(N_BSLD + P)$ |  $O(N_BSLD + P)$ |  $O(N_BSD + P)$
>  Differentiable Laplace | $O(N_BSLD + P)$ | $O(N_BSLD^2 + P)$ | $O(N_BSLD^2 + P)$
>
> The memory complexity of Differentiable Laplace scales quadratically with network-width, which can be prohibitively expensive. In trying to use their code-base on the small-scale CNN for our MNIST experiments, we ran out of memory on an RTX3080 10GB GPU.
>
> ### Practical scalability
> Big-$O$ notation hides the constant factors that can be important in practice. We report relative empirical time increase with respect to a regular parameter update during standard training for a single iteration in the table below. As with Immer et al. (2022), we measure the time required to update parameters and hyper-parameters separately. For Partitioned Networks and Diff. Laplace (as opposed to Augerino), these updates happen separately. Since we currently cannot run the code-base of (Immer et al., 2022) for Differentiable Laplace, we reference the measurements (relative to standard network training baseline) from their appendix and use Augerino in our implementation as a common denominator to compare timing-results between the three methods.
>
>
> | Method                          |Mnist CNN | ResNet8 | ResNet14
> ---------------------------------| :--- | :--- | :--- |
> Augerino | $\times1$ | $\times1$ |$\times1$ |
>  Diff. Laplace param.            | $\times1$ | $\times1$ |$\times1$ |
>  Diff. Laplace hyperparam        | - | $\times 18.2$ | - |
>  Partitioned Networks param.     | $\times 1.08$ | $\times 1.17$ | $\times 1.21$
>  Partitioned Networks hyperparam | $\times 1.08$ | $\times 1.08$ | $\times 1.09$
>
> For more details, see Appendix C.2 of the revised submission.

---

> > ### Author Response · Authors · 2022-11-14
> > **Response to all reviewers [3 / 3]**
> >
> > ## Function space approximation
> > Several reviewers raised a concern regarding our statement that partitioning could lead to a reasonable approximation in the function space. This is merely an intuition that we have, rather than a rigorous statement. We have updated the text in the main submission to reflect that. Here we clarify the grounds for that intuition:
> >
> > Firstly, the mapping from neural network parameters to functions is not bijective, therefore, there could be several weight configurations leading to the desired target function and, for our purposes, it suffices to identify one. Secondly, neural networks are overparametrized, and many works have shown that they can be pruned significantly without affecting predictive performance [1], i.e., the pruned networks can approximate dense networks quite well in function space. As a result, it is not unreasonable to expect to be able to approximate a specific predictive function with only a subset of the parameters, e.g. a subnetwork arising from our partitioned approximation. Finally, "scaling laws" in deep learning indicate that the benefit of having more parameters becomes apparent mostly for larger dataset sizes [2], therefore, it is also reasonable to expect that subnetworks "seeing" more data require more learnable parameters.
> >
> > ---
> >
> > Please let us know if you have any other comments, feedback, concerns, or anything else we can elaborate on!
> >
> > [1] Frankle, Jonathan, et al. "The lottery ticket hypothesis: finding sparse, trainable neural networks."  ICLR 2019. https://openreview.net/forum?id=rJl-b3RcF7
> >
> > [2] Kaplan, Jared, et al. "Scaling laws for neural language models", arXiv preprint arXiv:2001.08361, 2020. https://arxiv.org/abs/2001.08361

---

### Author Response · Authors · 2022-11-24
**Response to All Reviewers - Post-rebuttal Revision Follow-up**

We would like to update the reviewers on the changes in the final rebuttal revision of the paper that we hope will address the remaining comments and questions. Thank you again for pointing out where we can improve and clarify our paper.

## How good are the hyperparameters found?
Some reviewers asked how good the hyperparameters found by partitioned networks are.
We performed additional experiments during the rebuttal period to explore this question.
These experiments confirm that **1)** using the hyperparameters found with partitioned networks to retrain a "standard" neural network yields similar or better performance compared to that of the partitioned network, and **2)** the performance of the retrained network is among the best achieved through a fairly extensive random search of the hyperparameter space.
This indicates that the hyperparameters found by partitioned networks are indeed a good set of hyperparameters that translate well to standard neural networks, at least in the context of data augmentation learning.
We added a new section "How good are the hyperparameters found?" (Appendix F) to the appendix to showcase these experiments.

## Sensitivity to Partitioning
In the latest revision, we also further expanded the section on the sensitivity of the results to the partitioning scheme.
In Appendix E, we now include an ablation over the number of chunks, the proportion of datapoints in each chunk, and the proportion of parameters assigned to each chunk on the affine data augmentation learning task on MNIST and RotMNIST.
We report the final test set performance on those datasets.
Additionally, to gauge whether the hyperparameters found by partitioned networks with different partitioning schemes are good hyperparameters, we retrain standard models on this task using the final set of hyperparameters found by each respective partitioned run.
The results indicate that there are minor variations in final test performance depending on the partitioning scheme chosen, but the final set of hyperparameters found appears to be similarly good when used to retrain a standard neural network model.

---

### Decision · Program_Chairs · 2023-01-20

**Decision:**

Accept: poster

**Justification For Why Not Higher Score:**

* Missing an experimental validation at a larger-scale to increase its practical significance.
* Missing more HPO-related comparisons (techniques & settings) to demonstrate the method is an actual alternative for HPO problems
* Connection with marginal likelihood could be improved and beefed up.

**Justification For Why Not Lower Score:**

* Original idea and partitioning scheme.
* Well-written and well-presented manuscript.
* Relevant/well-motivated applications to federated learning.

**Metareview: Summary, Strengths And Weaknesses:**

The reviewers and meta reviewer all carefully checked and discussed the rebuttal. They thank the authors for their response and their efforts during the rebuttal phase.
The rebuttal notably strengthens some aspects of the experimental validation of the work (quality of the found hyperparameters, sensitivity analysis w.r.t. partitioning, measure of training overhead, more detailed related work) and confirms the interesting, original and sound contributions of the paper, which should inform future research (as acknowledged by the reviewers).

However, there are some important aspects that have not been fully addressed during the rebuttal. _As a result, the reviewers and the meta reviewer are weakly inclined to accept the paper._
In particular, the authors are urged to carefully update their final manuscript with the following points:

(i) The discussion about the connection with the marginal likelihood should be further developed and should be made more formal. Right now, several arguments are still intuitive. Similarly, the notation of the sections where the discussion takes place (Sec. 2 and 3) needs to be improved. For example, the prior over function (f) is not never really discussed, and the mix between w and f is confusing, especially because we only deal with parametric models (w) in the paper.
It would be important to add a complete example of what happens when the model is very simple (say linear) and all the derivations (e.g., assuming Gaussian distributions) can be exactly computed and easily presented.

(ii) Right now, the experiments operate at an arguably small/moderate scale, for both the model and the datasets. It would be important to probe the behavior of the proposed approach on larger scales (e.g., ImageNet). Indeed, this is at a larger scale that hyperparameter tuning becomes a real bottleneck.

(iii) To fully prove its generalizability, it would be important to complementarity report the performance of the proposed approach for the tuning of more “standard” hyperparameters such as L2 regularization and dropout rate.

(iv)  Several claims in the paper, e.g., “Compared to other non-Bayesian methods like Benton et al. (2020), our approach is broadly applicable to any hierarchical modelling setup” or the fact that the proposed approach is a mature alternative to HPO, need to be clarified and/or toned down. For the latter example claim, there are only a few HPO baselines/settings considered in the experiments.

If the paper was submitted to a journal, it would be accepted conditioned on those key changes, the meta-reviewer thus expects all those changes to be carefully implemented.


**Note From Pc:**

if the above contains the word "oral" or "spotlight" please see: "oral" presentation means -> notable-top-5% and "spotlight" means -> notable-top-25%. As stated in our emails, we are disassociating presentation type from AC recommendations

**Summary Of Ac-Reviewer Meeting:**

The meta reviewer and two (out of the four) reviewers met on 12/7.
In the meeting, we quickly converged to a “weak accept“ decision by collectively articulating the first three points listed in the meta review, namely:

(i) Strengthening of the connection between the marginal likelihood & the approach + Simple example needed.

(ii) Scale of the experiments.

(iii) Generalizability of the approach.

One of the reviewers called out examples of claims that were still not clear and/or not substantiated after the rebuttal, leading to the point (iv).

---

> ### Author Response · Authors · 2023-03-02
> **Camera ready revision**
>
> We would like to thank the AC and the reviewers for accepting our work for the conference. We also highly appreciate the additional pointers for further improving our work. We addressed all of them for the camera-ready version. More specifically:
>
> i)	We revised appendix A, and now show how our approximation at Eq. 2 lower-bounds the marginal likelihood of a finite-dimensional parametric model, i.e., it considers the weight space $w$ instead of the function space $f$ under some assumptions. We also added appendix B, showing how partitioned networks can be used to approximate the marginal likelihood; they can be understood as specific variational approximations to the true posteriors after seeing specific chunks of data. We hope that this is enough to clear any confusion.
>
> ii)	To demonstrate the scalability of our method, we added additional experiments on TinyImagenet with a ResNet-50 architecture, where, similarly to CIFAR10, we show that partitioned networks can optimize hyperparameters successfully, hence improving upon the baseline.
>
> iii)	We would like to highlight that for our federated learning experiments we introduced dropout to the neural network architectures we considered. We then used partitioned networks to optimize, besides the affine augmentation parameters, the dropout rates as well.
>
> iv)	We revised some of our claims appropriately. We would also like to point out that we do not mention in the paper that our work is a mature alternative to HPO; to become such an alternative, more research is needed in order to improve upon the limitations we discuss in appendix H.